# Elucidating essential kinases of endothelin signalling by logic modelling of phosphoproteomics data

Alexander Schäfer[1,*] (ID), Enio Gjerga[2] (ID), Richard WD Welford[3], Imke Renz[3], Francois Lehembre[3], Peter MA Groenen[3], Julio Saez-Rodriguez[2,4] (ID), Ruedi Aebersold[1,5] (ID) & Matthias Gstaiger[1,6,**]

## Abstract

Endothelins (EDN) are peptide hormones that activate a GPCR signalling system and contribute to several diseases, including hypertension and cancer. Current knowledge about EDN signalling is fragmentary, and no systems level understanding is available. We investigated phosphoproteomic changes caused by endothelin B receptor (ENDRB) activation in the melanoma cell lines UACC257 and A2058 and built an integrated model of EDNRB signalling from the phosphoproteomics data. More than 5,000 unique phosphopeptides were quantified. EDN induced quantitative changes in more than 800 phosphopeptides, which were all strictly dependent on EDNRB. Activated kinases were identified based on high confidence EDN target sites and validated by Western blot. The data were combined with prior knowledge to construct the first comprehensive logic model of EDN signalling. Among the kinases predicted by the signalling model, AKT, JNK, PKC and AMP could be functionally linked to EDN-induced cell migration. The model contributes to the system-level understanding of the mechanisms underlying the pleiotropic effects of EDN signalling and supports the rational selection of kinase inhibitors for combination treatments with EDN receptor antagonists.

**Keywords** endothelin; GPCR; melanoma; molecular modelling; phosphoproteomics

**Subject Categories** Post-translational Modifications, Proteolysis & Proteomics; Signal Transduction; Proteomics

**Mol Syst Biol. (2019) 15: e8828**

## Introduction

Endothelin is the most potent and longest lasting vasoconstrictor in human vasculature. The three EDN isoforms endothelin-1, endothelin-2 and endothelin-3 are paracrine and autocrine signalling peptides, each consisting of 21 amino acids. Their receptors EDNRA and EDNRB are class A GPCRs and differ in their expression patterns and physiological roles (Barton & Yanagisawa, 2008). The three EDN isoforms bind both receptors with sub-nanomolar affinities and are redundant in their function, with the exception of endothelin-3 which has ~ 100-fold lower affinity for EDNRA (Kedzierski & Yanagisawa, 2001).

Intense study of the EDN system has established its physiological functions in regulating vascular tone, the cardiopulmonary system and salt homeostasis in the kidney (Kedzierski & Yanagisawa, 2001). EDN is also a crucial factor in neural crest and melanocyte development (Saldana-Caboverde & Kos, 2010). Aberrations in the EDN system have been associated with a range of pathologies, and consequently, EDN receptor inhibition has proven beneficial for pulmonary arterial hypertension, congestive heart failure and kidney failure (Barton & Yanagisawa, 2008). EDN receptor blockers have become one of the main treatment options for managing pulmonary arterial hypertension (Galie et al, 2013).

The EDN system has also been recognised to influence a range of basic cellular functions related to cancerogenesis. Most notably, it activates cell proliferation, inhibits apoptosis, induces extracellular matrix remodelling (Nelson et al, 2003) as well as angiogenesis (Spinella et al, 2002) and activates cell migration (Rosano et al, 2006). Furthermore, EDN autocrine or paracrine activation has been described in ovarian, prostate, colon, breast, bladder and lung cancer. Increased EDN pathway activation is associated with tumour malignancy (Rosano et al, 2013). Despite convincing preclinical evidence for a range of different tumour types (reviewed in Nelson et al, 2003), most clinical studies in the field of oncology

1 Department of Biology, Institute of Molecular Systems Biology, ETH Zurich, Zurich, Switzerland
2 Faculty of Medicine, Joint Research Centre for Computational Biomedicine (JRC-COMBINE), RWTH Aachen University, Aachen, Germany
3 Idorsia Pharmaceuticals, Allschwil, Switzerland
4 Faculty of Medicine, Institute for Computational Biomedicine, Heidelberg University Hospital, Bioquant, Heidelberg University, Heidelberg, Germany
5 Faculty of Science, University of Zürich, Zürich, Switzerland
6 Competence Center Personalized Medicine UZH/ETH, Zürich, Switzerland
*Corresponding author. Tel: +41 44 633 25 98; E-mail: schaefer@imsb.biol.ethz.ch
**Corresponding author. Tel: +41 44 633 34 49; E-mail: matthias.gstaiger@imsb.biol.ethz.ch

Molecular Systems Biology   15: e8828 | 2019   1 of 19

to date have not shown significant benefit of EDN inhibition (Rosano *et al*, 2013), suggesting that a deeper molecular and system-oriented understanding of EDN signalling is required to devise more promising, knowledge-based treatment strategies, e.g. through matching specific drug combinations with patient subgroups.

A case in point is EDN signalling in melanoma. Despite new treatment modalities, cancer death rates in advanced melanoma remain high, with a 5-year survival rate of 20% (Siegel *et al*, 2018). A strong induction of EDNRB has been described in a subgroup of melanoma patients (Asundi *et al*, 2011), and EDNRB expression has been found to correlate with melanoma malignancy (Demunter *et al*, 2001). Functional studies have shown that EDNRB inhibition leads to melanoma cell death in culture (Asundi *et al*, 2011) and reduces melanoma growth in mice (Lahav *et al*, 1999) through induction of apoptosis (Lahav *et al*, 2004). Melanoma invasiveness, another hallmark of cancer, has been linked to aberrant EDN signalling: overexpression of EDNRB in melanoma mouse models increases metastasis while its inhibition has the reverse effect (Cruz-Munoz *et al*, 2012). The most prevalent oncogenic driver mutation in melanoma, BRAF (V600E), activates the MAPK pathway. Treatment with mutant BRAF targeting drugs leads to dramatic responses and regression, but drug resistance develops in most cases and patients' relapse (Fedorenko *et al*, 2015). A number of different mechanisms have been implicated in the relapse, including EDN signalling (Smith *et al*, 2017). Two recent studies have shown that EDNRB inhibition can counter BRAF resistance development in mouse models, significantly increasing survival (Smith *et al*, 2017; Renz *et al*, manuscript in preparation). Mechanistic details of this interaction between MAPK pathway inhibition and EDN signalling have remained elusive.

Despite the profound and well-documented influence of EDN signalling on physiology and disease development, current knowledge of the underlying signalling mechanisms is fragmentary. The consensus model of ENDRA signalling integrates evidence from more than 20 studies performed in various model systems from five different species (Rosano *et al*, 2013). These studies relied on antibodies for targeted analysis of established signalling pathways, highlighting well-known molecular circuits. The EDN signalling pathway is probably much more complex than presently appreciated, as GCPRs are known to interface with an extensive kinase network through heterotrimeric G proteins and arrestin (O'Hayre *et al*, 2014). The existing EDN consensus model is heterogeneous, does not take into account the systemic character of the phosphorylation response and contains conflicting evidence for some important mechanistic aspects, e.g. MAPK activation (Imokawa *et al*, 2000; Cramer *et al*, 2001).

As a consequence, a comprehensive map of EDN-mediated protein phosphorylation, EDN-responsive kinases and the connecting network structure is currently lacking. To address this scientific need, phosphoproteomics was used to generate the first comprehensive network model of EDN signalling, derived from a single system. EDNRB signalling in melanoma cells was chosen, as it has a defined role in the development and homeostasis of this cell type and its dysregulation has been implicated in melanoma pathogenesis (Saldana-Caboverde & Kos, 2010).

The melanoma cell lines UACC257 and A2058 were selected for the time-resolved phosphoproteomic study because (i) they have no measurable EDNRA expression (Renz *et al*, manuscript in preparation), so that EDN signalling is mediated purely through EDNRB, (ii) a CRISPR/Cas9 EDNRB knockout was available, and (iii) the system is of broad medical interest.

We systematically quantified time resolved, EDNRB activation-induced protein phosphorylation changes. We related reproducibly observed phosphopeptide patterns to upstream kinases and established the first large-scale network model of EDNRB signalling based on prior knowledge of kinase–substrate (K-S) relationships and protein–protein interactions (PPI) trained with the phosphoproteomics data. Finally, the functional relevance of four kinases, predicted to be central nodes in the network, was demonstrated by testing the effect of kinase inhibition on EDN-induced cell migration. Overall, this study and the resulting model provide a deeper understanding of the molecular network effecting EDN signalling and may enhance the pharmacological exploitation of this clinically relevant pathway.

# Results

### Phosphoproteomic analysis of EDNRB activation in melanoma cells

#### Quantitative and time-resolved phosphoproteomic study of EDNRB signalling

In a first step, experimental conditions to study EDNRB were optimised. To establish the timeframe for EDNRB signalling, intracellular $Ca^{2+}$ release and two known EDNRB target phosphosites AKT S473 (Liu *et al*, 2003) and CREB S133 (Schinelli *et al*, 2001) were measured in both parental cell lines and the respective EDNRB-KO derivatives. EDN caused a rapid release of $Ca^{2+}$ in UACC257 and A2058 cells, which peaked after a few seconds and subsided over approximately 10 min (Fig 1A). The response in UACC257 cells was more sustained than in A2058. EDNRB knockout abrogated the calcium transient in both cell lines (Fig 1A). On the phosphorylation level, EDN caused a transient induction of CREB S133 phosphorylation (Fig 1B) with a maximum at 10 min in UACC257 and 2 min in A2058. In contrast, AKT S473 phosphorylation was induced after 10 min and plateaued until 90 min (Fig 1B) in both cell lines. Both phosphorylation events were dependent upon the expression of EDNRB in UACC257 (Fig 1B). The observed time dependencies indicate that EDNRB signalling consists of both, rapid transient and sustained responses in these cell lines.

The phosphoproteomic experiment design was derived from the optimised stimulation conditions. Samples were processed according to the digestion and phospho-enrichment protocol and LC-MS/MS data-dependent acquisition workflow described in the Materials and Methods section. Two time-resolved EDN phosphoproteomic data sets with the time points 2, 10, 30, 60 and 90 min were generated: one for UACC257 WT and UACC257 EDNRB-KO (Fig 1C), and one for A2058 WT. For UACC257 cells (WT and EDNRB-KO), a total of 5,240 unique phosphopeptides (Table EV1) and 5,172 non-redundant phosphosites (Table EV1) were quantified. For the A2058 cell line, 5,832 unique phosphopeptides (Table EV2) and 5,568 non-redundant phosphosites (Table EV2) were quantified. These numbers are already corrected for phosphate localisation, which was controlled with LuciPHOr (Fermin *et al*, 2013) at 1% false

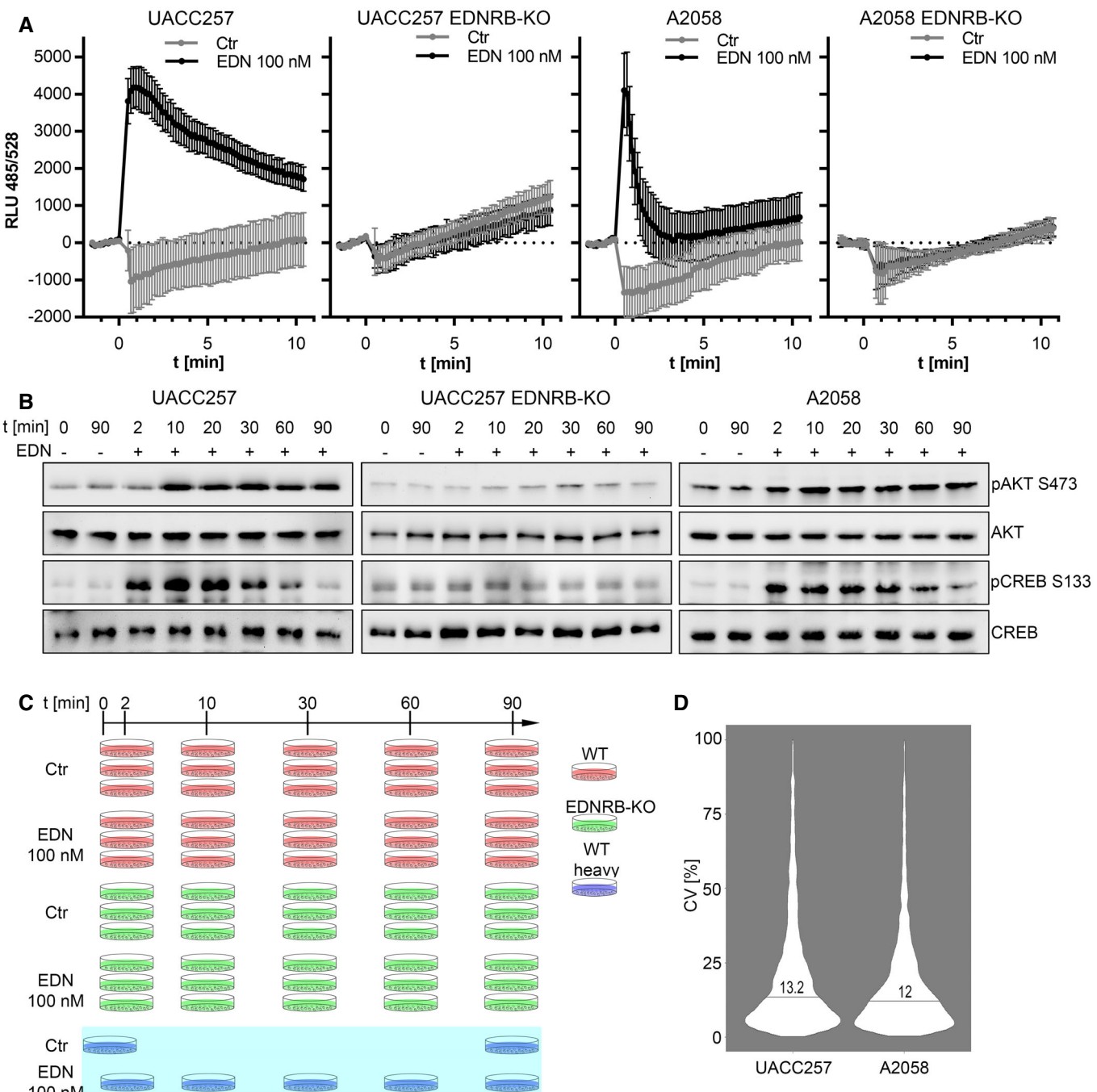

**Figure 1. UACC257 and A2058 melanoma cells are a model system to globally study EDNRB signalling.**

A   Kinetics of intracellular Ca²⁺ after EDN stimulation. UACC257 and A2058 (WT and ENDRB-KO) were seeded on 96-well plates, loaded with Fluo-4 dye and stimulated with 100 nM EDN or PBS. Fluo-4 fluorescence was monitored over 10 min ($n = 6$). Representative example of three independent experiments. The error bars indicate ± standard deviation.

B   Time course of AKT and CREB phosphorylation. Melanoma cell lines were stimulated with 100 nM EDN for 2–90 min. AKT and CREB phosphorylation and expression were evaluated by Western blot. Representative example of three independent experiments.

C   Experiment design for the UACC257 phosphoproteomic study, showing time points, cell line derivatives, replicate structure and GIST composition. The GIST was generated by pooling all heavy SILAC plates (blue) and spiked into all UACC257 (red) and UACC257 EDNRB-KO (green) samples. The A2058 cell line was also analysed in biological triplicates and a GIST but without EDNRB-KO cells.

D   High reproducibility in the UACC257 and A2058 data sets. CV distributions for triplicates were calculated and are provided in Fig EV1D. All CV values were grouped according to cell line to represent data set-specific CV distributions. Numbers and lines indicate the median.

localisation rate (FLR; see "Data analysis to generate phosphoproteomic data sets"). To achieve consistent quantification across WT and KO cells, multiple time points and mock-stimulated controls, a SILAC labelled global internal standard (GIST) spike-in approach was devised (Fig 1C). Heavy labelled cells were subjected to most experimental conditions and pooled to generate the GIST, which was spiked into the non-SILAC labelled lysates as an internal reference sample.

Reproducibility was assessed by comparing the CV distributions for all 30 biological triplicates in this study. CV distributions of triplicates were combined into one distribution per data set to assess overall reproducibility (Fig 1D), with median CV values 13.2 and 12.0% for UACC257 and A2058, respectively. A detailed analysis with individual CV distributions in all triplicates is provided in Fig EV1D and showed median CVs ranging from 11.1 to 14.6%. These values were in agreement with a previous benchmarking of the workflow (see "Data analysis to generate phosphoproteomic data sets").

The descriptive characteristics of the generated phosphopeptide data sets were highly similar to previously published $TiO_2$ enrichment data (Olsen *et al*, 2006). Most [75% (UACC257) and 78% (A2058)] phosphopeptides were singly phosphorylated, while the rest were mostly doubly phosphorylated (Fig EV1E). Phosphoserine was most frequently observed (82 and 85% for UACC257 and A2058, respectively), while phosphotyrosine residues were rare (2–3%; Fig EV1F).

The GIST approach has the key advantage that the analysis structure of a label-free experiment can be used, generating a data matrix in which all values for the same peptide can be compared directly, while retaining the high quantitative reproducibility and robustness of SILAC in the lengthy tryptic digest and phosphopeptide enrichment procedure. However, the GIST approach shares a major drawback of SILAC—a reduction of identification rates due to increased sample complexity.

The extensive and high-quality phosphoproteomic data sets in UACC257 and A2058 were used as the basis for all subsequent analyses.

### ENDRB activation leads to robust phosphorylation changes affecting hundreds of target phosphosites in UACC257 and A2058 cells

Effects of EDNRB stimulation were assessed on the level of phosphorylation patterns and individual phosphopeptides to determine EDNRB-dependent phosphorylation events. Principal component analysis of the phosphoproteomic data set for UACC257 indicated a clear separation of experimental groups (Fig 2A). PC1 separated WT from EDNRB-KO samples, while PC2 separated EDN-stimulated WT from mock-stimulated WT samples. EDNRB-KO samples were not separated by PC2 according to their EDN stimulation status. It was even possible to discern a trend for stimulation duration in the WT EDN-stimulated group along PC2 (Fig 2A). A similar separation of samples according to the presence and activation of EDNRB was obtained using hierarchical clustering (Fig EV2A). PCA of phosphopeptide MS1 intensities for the A2058 data set also showed a clear separation of control and EDN-stimulated samples as well as separation of samples according to stimulation duration (Fig EV2B). The PCA analysis demonstrated that the expression and activation of EDNRB translated into two different alterations of the phosphorylation pattern. In addition, it supported the quality of the data set by showing that biological replicates clustered together.

Next, differentially abundant phosphopeptides were identified by statistical testing with multiple testing correction at $q < 0.1$ and fold change of > 1.5 up or down. Using these criteria, between 200 and 500 EDN target peptides were discovered for each time point in both UACC257 and A2058 (Table 1). The same analysis was performed for the EDNRB-KO data set, with only two peptides crossing the threshold at a single time point (Table 1). This demonstrates that all observed changes in the UACC257 cell line were a consequence of EDNRB activation and were strictly dependent on this receptor. In total, 918 (UACC257) and 665 (A2058) EDNRB target peptides were identified (Table 1).

To exclude that differential phosphopeptide abundances were caused by changes in protein abundance rather than changes in phosphorylation, quantitative protein profiles were generated by DIA-SWATH on protein extracts from UACC257 and A2058 cells that were either mock or EDN stimulated for 90 min. The latest (90 min) time point was chosen because differential protein expression was expected to be particularly pronounced. Almost 3,000 proteins were quantified in each cell line by proteotypic peptides. No proteins with differential abundance between EDN and mock-treated cells were identified at $q < 0.1$ with no FC cut-off (Table 1 and Table EV3). However, only half of the almost 2,000 phosphoproteins from which phosphopeptides were quantified after phosphopeptide enrichment were included in the protein abundance data set (Table 1). Nevertheless, the analysis indicates that most of the observed changes on the phosphorylation level were likely to be caused by differential phosphorylation.

Next, the EDN response between the two cell lines was compared. The purpose was to establish a core set of EDN-responsive target peptides which were affected consistently in both cell lines. On the identification level, the overlap of phosphoproteins identified between cell lines was 57.6% (1,433), and for phosphopeptides identified, the overlap was 38.2% (2,899; Fig 2B). Comparing EDN-regulated phosphopeptides, 375 target peptides (19.6%) were shared at $q < 0.1$. When the FC cut-off was applied, this intersect was reduced to 242 (18%; Fig 2B). After removal of phosphopeptides which were only identified in one cell line, the 242 shared target peptides accounted for 33.6% of all EDN target peptides. The time–response curves of the 242 shared target phosphopeptides were highly similar between the cell lines (Fig 2C). It was also apparent that EDN-induced phosphorylation changes up to 90-min stimulation were predominantly increases (green in Fig 2C).

The comparison identified a core set of 242 shared target peptides, which were induced in both cell lines with similar kinetics, but also a significant difference in the overall response pattern on the level of individual phosphosites.

The data sets contained many EDN target phosphosites without functional annotation, but also some very well-established sites like ERK1 T202/Y204 and novel EDN-responsive phosphorylations on established kinases like MEK2 and ROCK2 (Fig EV2C).

## Systems biology analysis of EDNRB-mediated phosphoproteomic changes

### EDNRB-targeted phosphosites delineate a discrete subset of cellular processes

To gain an overview of the functional implications of EDN-induced phosphorylation, pathway enrichment analysis was performed.

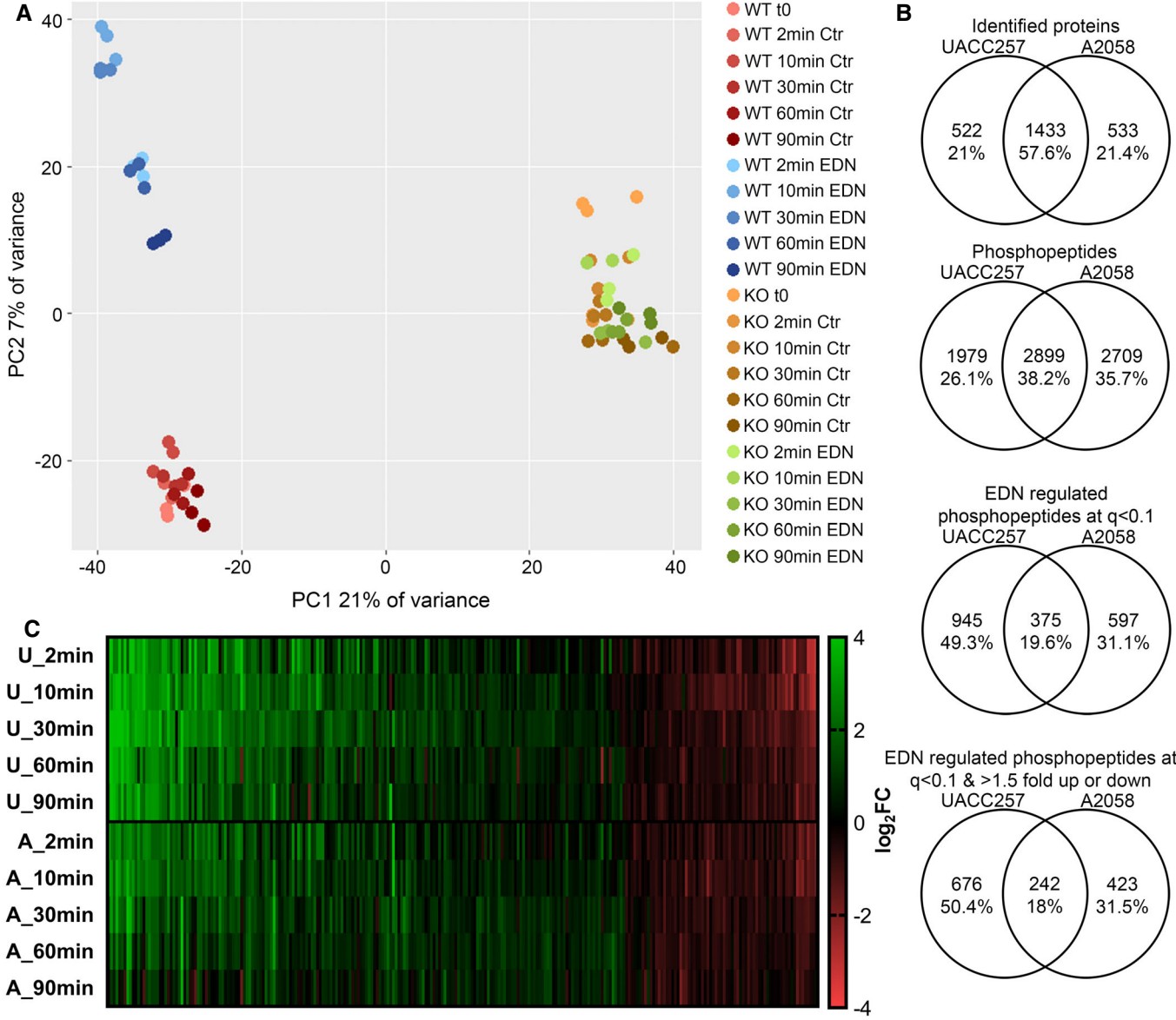

**Figure 2. EDN elicits a robust phosphorylation response, which is partly shared between melanoma cell lines.**

A  Phosphopeptide MS1 intensities in UACC257 samples were analysed by PCA. Colours indicate experimental groups, and shading is mapped to time.

B  Overlap between UACC257 and A2058 data sets. From top to bottom: Identified proteins, identified phosphopeptides, EDN-regulated phosphopeptides at $q < 0.1$, and $q < 0.1$ & FC > 1.5 up or down.

C  Comparison of the kinetics of the 242 shared target phosphopeptides. Log$_2$ FC of EDN/Ctr is plotted from top to bottom for each phosphopeptide for both cell lines with increases shown in green and decreases shown in red.

Each time point in each cell line was analysed separately using IPA (Fig 3). Separate analyses yielded similar list of terms which were compiled into an integrated comparison shown in Fig 3. For clarity of presentation, lower order terms are not shown and a complete list of terms is provided in Table EV4. Enriched processes and pathways showed strong similarity between cell lines, including a similar temporal pattern. The largest increase in enriched terms occurred between 2 and 10 min, following the dynamics of the number of regulated phosphopeptides. The significance and number of terms decreased after 30-min stimulation.

Significant similarity of the EDNRB targets with known GPCR signalling pathways (e.g. Thrombin, CXCR4) was found, but the targets also shared similarity with multiple receptor tyrosine kinase (e.g. VEGF, HGF) and cytokine receptor signalling pathways (e.g. IL4, EPO; Fig 3A). Furthermore, EDN target sites were involved in cytoskeleton-associated signalling (e.g. integrin signalling) or belonged to established kinase cascades (Fig 3A). The latter contained known GPCR-responsive kinases (PKC, ROCK) but also kinase not commonly associated with GPCR signalling (e.g. p70S6K, PAK or stress-activated MAPKs). For cellular processes, EDN target

**Table 1. Summary of EDN-induced phosphorylation and expression changes.**

| UACC257—EDN target phosphopeptides | | | |
|---|---|---|---|
| Time point | WT | EDNRB$^{-/-}$ | Union |
| 2 min | 210 | 0 | 918 |
| 10 min | 505 | 2 | |
| 30 min | 496 | 0 | |
| 60 min | 429 | 0 | |
| 90 min | 389 | 0 | |

| A2058—EDN target phosphopeptides | | | |
|---|---|---|---|
| Time point | WT | | Union |
| 2 min | 242 | | 665 |
| 10 min | 233 | | |
| 20 min | 254 | | |
| 30 min | 275 | | |
| 60 min | 289 | | |
| 90 min | 251 | | |

| | | UACC257 | A2058 |
|---|---|---|---|
| SWATH protein data set | Protein IDs | 2808 | 2965 |
| | $q < 0.1$ | 0 | 0 |
| Phosphoproteomic data set | Phosphoproteins | 1976 | 1986 |
| | Overlap with SWATH | 843 (43%) | 864 (43%) |

From top to bottom: The number of regulated phosphopeptides ($q < 0.1$ & > 1.5 FC up or down) per time point for UACC257 WT and EDNRB-KO cells and A2058 cells. The lower section shows the outcome of the SWATH analysis after 90-min EDN stimulation: identified and differentially expressed proteins for both cells lines and their overlap with the proteins from which the phosphopeptides were derived.

sites were strongly enriched for processes involved in cell migration and motility as well as for functions related to the organisation of the cytoskeleton and cell morphology (Fig 3B). Smaller groups of terms showed enrichment for apoptosis regulation, intercellular communication and protein synthesis (Fig 3B).

### Highly similar sets of kinases are activated through EDNRB in UACC257 and A2058 cells

One effective way to obtain a better understanding of the complex phosphorylation patterns in response to EDNRB activation was to identify activated kinases as the central elements organising the signalling network. Two approaches were used to infer kinase activation: the first was based on experimentally validated kinase–substrate (K-S) relationships using the tool PHOXTRACK (Weidner et al, 2014). Since the overlap of experimentally validated K-S relationships and the phosphoproteomic data set was limited to a few dozen known target sites, a second approach based on NetworKIN (Linding et al, 2008) predicted K-S relationships was used to identify kinase activation with a higher number of annotated sites. The rationale for two prediction approaches based on different data sources was to generate an inclusive EDN target kinase list, which was then used as a broader basis for validation experiments.

Combining results for different time points and both cell lines, PHOXTRACK identified a total of 34 putative EDN target kinases

(Table 2). Redundancy in this list was due to identification of numerous kinase isoforms. Almost all target kinases were found to be activated (positive values in Table 2) with the exception of CDK isoforms and PDHK. No strong time dependency was observed for kinase activation. The largest increase in the number of activated kinases was observed between the 2-min and the 10-min time point. PHOXTRACK identified similar sets of kinases for both cell lines (Table 2).

Activation of the PI3K-AKT-p70S6 axis, already inferred from the first experiments (Fig 1B) and activation of the PKC axis, expected to occur based on the calcium release (Fig 1A), were found. GPCR-associated kinases PKA and ROCK were also identified. A major fraction of the identified kinases was involved in the MAPK pathways leading to activation of ERK as well as activation of the stress-regulated MAP kinases p38 and JNK. An unexpected finding was the activation of CK2, RSK, AMPK and PAK as well as inhibition of CDKs.

For NetworKIN-based predictions, target response profiles were first grouped by c-means clustering and enrichment analysis was performed for the different response kinetics in UACC257 (Fig 4A) and A2058 cells (Fig 4B). Kinase activations were inferred from an overrepresentation of their substrates in a cluster. In general, this meant that the substrates of an enriched kinase only accounted for a fraction of all curves in that cluster. A full list with target numbers can be found in Appendix Table S1.

Separation into six clusters yielded highly similar response kinetics for both cell lines. Clusters 1–4 contained phosphorylations with increased abundance and maxima at 2, 10, 30 and 60–90 min, respectively. Cluster 1 and 2 targets were strongly enriched for CaMKII targets, a kinase with a small number of targets in the prediction based on validated substrates. Cluster 3 showed the strongest signature for PAK activation in both cell lines. Activation of the linked kinases AKT and p70S6K was evident for clusters 2 and 3. While p70S6K appeared in both cell lines in cluster 3, AKT appeared in UACC257 in cluster 2 but in A2058 in cluster 3. A similar difference in kinetics was observed for the MAPK pathway. MEK activation was seen in UACC257 for sites peaking at 10 min (cluster 2) but appeared in A2058 after 60 min (cluster 4). ERK activation was then observed in UACC257 at 60 min (cluster 4), but was absent for A2058. PKC activation only appeared in clusters 1 and 2 for A2058, but was absent for UACC257. This was probably due to PKC target sites being more evenly distributed over clusters 1–4 in UACC257. Clusters 5 and 6 contained peptides with reduced phosphorylation with fast (cluster 5) and sustained (cluster 6) behaviour. GSK3-β was enriched in cluster 5 and is known to be inhibited by S9 phosphorylation downstream of different kinases, whose activation occurred at earlier time points [e.g. AKT (Salas et al, 2003)]. GSK3-β S9 phosphorylation was induced by EDN in A2058 but the site was not contained in the UACC257 data set. On the other hand, the fast drop in abundance suggested a more active mechanism, possibly activation of phosphatases. Cluster 6 had a weak signature for CDK targets, which was only significant in A2058.

The strongest signature for activated kinases identified by the NetworKIN approach was CaMKII. This kinase is part of a complex which is directly activated by $Ca^{2+}$. Comparison of the kinetics of 217 predicted CaMKII sites in both cell lines with the $Ca^{2+}$ release kinetics showed similar curves on different time scales (Fig 4C). CaMKII activity seems to relay the ~ 10-min calcium transient onto

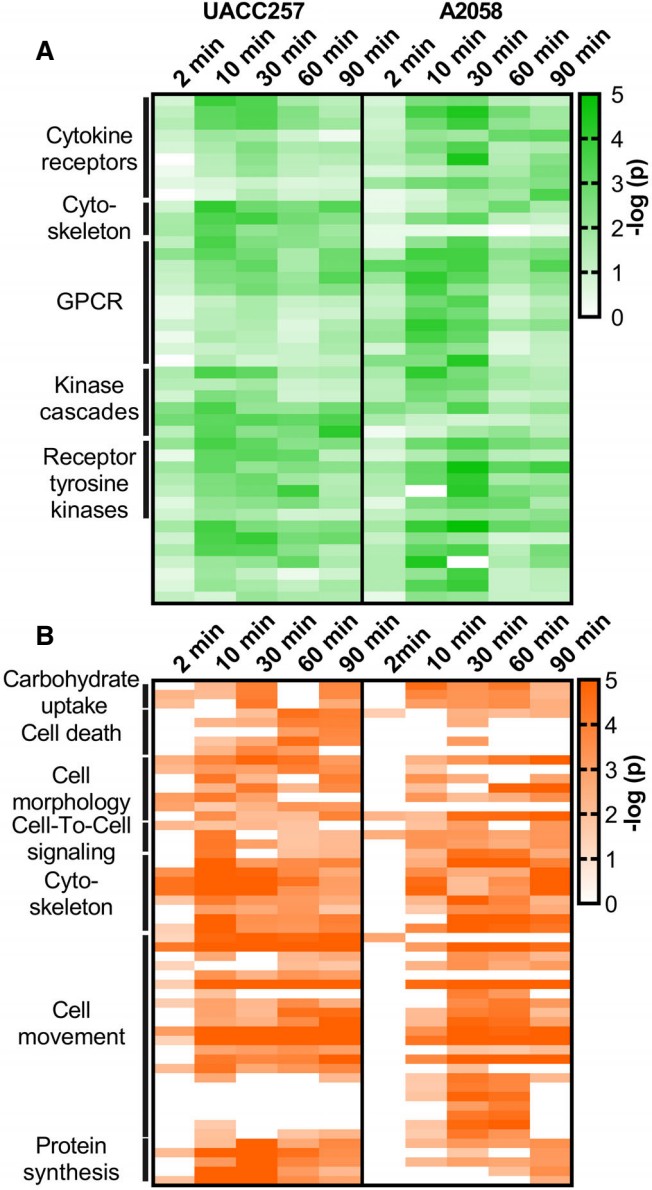

**Figure 3. EDN-activated substrates are associated with a defined set of cellular functions and canonical signalling pathways.**

A, B   EDN phosphoproteomic data were analysed in IPA. Significance of term enrichment is indicated through shading as —log (*P*-value). Regulated pathways belonging to the classes "Canonical Pathways" (A) and "Disease and Biofunction" (B) with significant enrichment (*P* < 10⁻³) are shown for UACC257 and A2058. Regulated terms were grouped according to highest order ontology terms, which are shown on the left.

a ~ 90-min phosphorylation pattern. UACC257 showed a more protracted response for both Ca²⁺ activity and CaMKII activity than A2058 (Fig 4C).

Combining the PHOXTRACK and NetworKIN identified kinases, and grouping their isoforms resulted in an inclusive list of 18 putative EDN target kinases. Seven kinases were identified by both approaches (AKT, PKC, PAK, p70S6K, MEK, ERK and CDKs). Nine kinases were only identified by PHOXTRACK (CKII, p38, JNK, RSK,

ROCK, MAPKAPK2, PKA, AMPK and RAF), and two kinases were only identified by NetworKIN (PKD and CaMKII). The two kinase prediction approaches complemented each other. While PHOXTRACK found a higher number of EDN target kinases, NetworKIN contributed confirmatory evidence in many cases and added information about the kinetic patterns of kinase activation. Combination of both approaches allowed identification of a broader range of involved kinases for validation experiments.

### Leveraging prior knowledge and quantitative phosphoproteomic data to construct the first EDNRB signalling network in a single system

Kinase activation prediction identified the spectrum of EDN target kinases, as described in the previous section. It did not, however, allow inference of the relationship between the proposed kinases and the structure of the signalling network. Computational modelling was used to generate a logic network model which best explained the observed phosphorylation patterns in the context of existing knowledge. Logic models are constructed by selecting nodes and edges from an extensive prior knowledge network to find a structure optimally describing the observations. Such a network is better suited to infer key signalling nodes, regulatory relationships and to provide mechanistic insight how the EDN signal propagates from the receptor to its target sites. The modelling approach described here was performed in parallel to the analysis detailed above and did not integrate information from PCA, IPA or kinase activation predictions.

Prior knowledge about K-S relationships and PPI was used to assemble binary relationships into an optimised model, predicting network topology without the aid of generic pathway maps. To this end, an extensive K-S database compiled from literature and online databases (Turei *et al*, 2016) was employed. The background K-S network was complemented with a generic GPCR PPI network to enable linking of EDNRB to kinases. Using the PHONEMeS approach (Terfve *et al*, 2015), all linear paths linking EDNRB to the target sites identified by the phosphoproteomic analysis in the melanoma cells were collected and an optimised solution combining these paths into a network was calculated using Integer Linear Programming. To make best use of the temporal resolution of the phosphoproteomic data, models were generated in an evolving manner. A first model was generated for the 2-min time point. This model was then used as a starting point for the 10-min time point, extending the existing structure with new branches to include target phosphosites activated at the later time point, and the process was repeated until all time points were included.

The resulting signalling model for UACC257 cells predicted a network of intermediary proteins (blue nodes) that transmit the signal from EDNRB to its target sites (red nodes). Each edge was assigned an entry time into the network (edge colour). Edges that were added at later time points were generally farther away from the receptor. In this way, evolution of the signalling cascade could be visualised on a network level. The topology of the network suggests that the signalling system branch into five modules from the receptor (Fig 5). First, the arrestin module leads to MAPK cascade activation and accounts for many target sites phosphorylated by the effector MAPKs ERK1/2 p38 and JNK. The second module is smaller and involves the Ca²⁺-dependent kinases CaMKII and CaMKK2 and their

**Table 2. PHOXTRACK kinase enrichment shows that EDN activates similar sets of kinases in UACC257 and A2058.**

| Kinase | Sites | 2 min | 10 min | 30 min | 60 min | 90 min | Sites | 2 min | 10 min | 20 min | 30 min | 60 min | 90 min |
|---|---|---|---|---|---|---|---|---|---|---|---|---|---|
| AKT | 15 | 2.1 | 2.6 | 2.1 | 1.7 | 1.7 | 21 | 1.2 | 2.3 | 2.7 | 1.8 | 2.3 | 2.3 |
| PDK1 | 0 | NA | NA | NA | NA | NA | 5 | 1.7 | 1.6 | 1.4 | 1.7 | 1.4 | 2.0 |
| AMPK | 6 | 2.2 | 1.6 | 1.0 | 1.2 | 2.1 | 4 | 1.8 | 1.2 | 2.2 | 1.5 | 0.9 | 0.7 |
| CaMKII | 0 | NA | NA | NA | NA | NA | 3 | 2.0 | 2.0 | 1.7 | 2.0 | 1.9 | 1.2 |
| CDK1 | 24 | −1.2 | −1.8 | −1.0 | −1.5 | −2.3 | 70 | −2.3 | −1.5 | −1.4 | −1.5 | −1.6 | −1.2 |
| CDK12 | 7 | 1.5 | −1.7 | −1.0 | 1.2 | 0.9 | 5 | 1.7 | 1.2 | 0.9 | 1.3 | −1.2 | −1.0 |
| CDK5 | 8 | −1.1 | −1.5 | −1.8 | −1.4 | −1.6 | 8 | −0.9 | −1.5 | −1.2 | −1.7 | −1.0 | −0.7 |
| CDK2 | 90 | −1.2 | −1.5 | −1.3 | −1.0 | −1.6 | 110 | −2.0 | −1.8 | −1.8 | −1.9 | −1.9 | −1.3 |
| CKII | 42 | 1.1 | 1.4 | 1.6 | 1.7 | 1.0 | 53 | 1.3 | 1.7 | 1.8 | 1.8 | 0.9 | −1.1 |
| JNK1 | 8 | −1.2 | 1.2 | 1.7 | 1.5 | 1.1 | 8 | −0.9 | 1.0 | 1.1 | 1.4 | 0.9 | −0.9 |
| JNK2 | 3 | −2.1 | NA | 1.8 | 2.1 | NA | 3 | −2.0 | −0.8 | 1.3 | 1.4 | 1.2 | −1.7 |
| p38-α | 3 | −1.4 | −1.4 | 2.2 | 1.3 | 1.7 | 3 | −2.1 | −0.8 | 1.3 | 1.4 | 1.6 | −1.7 |
| p38-β | 3 | −2.1 | −2.1 | 1.9 | 1.8 | 1.7 | 0 | NA | NA | NA | NA | NA | NA |
| cRAF | 4 | 1.4 | 1.2 | 0.8 | 1.8 | 1.3 | 4 | 0.7 | −0.7 | −0.5 | 0.7 | −0.9 | 0.8 |
| MEK | 4 | 1.6 | 2.1 | 2.1 | 1.4 | 1.9 | 4 | −0.8 | 1.5 | 1.6 | 1.6 | 1.8 | 1.6 |
| ERK1 | 6 | 1.6 | 2.2 | 2.0 | 1.6 | 1.6 | 11 | −1.2 | 1.9 | 2.0 | 1.6 | 1.3 | 1.0 |
| ERK2 | 8 | 1.1 | 2.1 | 2.1 | 1.9 | 1.8 | 7 | 0.6 | 2.1 | 1.7 | 1.7 | 1.7 | 2.0 |
| MAPKAPK2 | 3 | 1.3 | 1.7 | 1.2 | 1.3 | 1.3 | 0 | NA | NA | NA | NA | NA | NA |
| mTOR | 16 | −1.2 | 1.7 | 2.2 | 1.4 | 1.1 | 18 | −1.1 | −1.1 | −1.2 | −1.3 | −1.0 | 1.3 |
| PAK1 | 11 | 1.1 | 1.5 | 1.8 | 2.3 | 2.0 | 11 | 0.9 | 0.6 | 0.9 | 1.5 | 1.8 | 1.1 |
| PAK2 | 9 | 1.9 | 2.2 | 2.3 | 2.2 | 2.2 | 8 | 0.8 | 1.9 | 2.0 | 2.2 | 2.1 | 2.4 |
| PKA | 22 | 1.5 | 2.5 | 2.5 | 1.9 | 2.0 | 26 | 2.3 | 1.6 | 2.2 | 2.8 | 2.1 | 1.4 |
| PKC-α | 9 | 3.2 | 3.2 | 2.3 | 2.0 | 1.8 | 12 | 2.9 | 3.0 | 2.8 | 2.0 | 2.6 | 1.0 |
| PKC-β | 4 | 1.7 | 1.7 | 1.0 | 1.4 | 1.4 | 4 | 1.3 | 0.8 | 0.8 | 1.0 | 1.3 | 1.2 |
| PKC-δ | 9 | 1.7 | 1.9 | 1.9 | 1.9 | 1.8 | 13 | 1.3 | 1.5 | 1.8 | 1.7 | 2.1 | 1.4 |
| nPKC-ε | 0 | NA | NA | NA | NA | NA | 5 | 1.4 | 1.6 | 1.7 | 1.7 | 1.6 | 1.4 |
| nPKC-μ | 8 | 2.0 | 1.6 | 1.4 | 2.4 | 2.3 | 3 | 2.1 | 2.2 | 2.2 | 2.2 | 2.2 | 2.1 |
| nPKC-η | 3 | 1.8 | NA | NA | 1.7 | 1.6 | 4 | 1.9 | 2.1 | 2.0 | 1.9 | 2.1 | 2.2 |
| ROCK1 | 3 | 1.4 | 1.3 | 1.1 | 1.9 | 1.4 | 3 | 1.2 | 1.2 | NA | 1.9 | 1.4 | 1.5 |
| ROCK2 | 4 | 1.8 | 1.7 | 1.5 | 2.1 | 1.5 | 4 | 1.5 | 1.2 | 1.5 | 2.1 | 1.6 | 1.7 |
| PDHK1/2 | 3 | −2.2 | −2.0 | −1.8 | −1.3 | −1.8 | 0 | NA | NA | NA | NA | NA | NA |
| S6K | 6 | 1.5 | 1.8 | 2.5 | 2.8 | 2.6 | 8 | 0.8 | 2.1 | 2.7 | 2.4 | 2.5 | 2.6 |
| RSK1 | 8 | 1.6 | 2.1 | 2.2 | 2.0 | 1.7 | 10 | 1.0 | 2.2 | 2.2 | 1.9 | 2.3 | 2.3 |
| RSK2 | 5 | 1.9 | 1.8 | 2.2 | 1.4 | 1.7 | 7 | 1.0 | 1.9 | 2.2 | 1.8 | 2.2 | 2.2 |

Kinase activation was predicted based on experimentally validated K-S relationships for each time point in both cell lines. The consensus table lists normalised enrichment values (NEV), positive for activation and negative for inhibition. The number of measured sites for the NEV calculation is given for each cell line. Grey shading indicates significance at $q < 0.1$. Kinases isoforms and kinases belonging to the same cascade were grouped manually.

downstream kinase AMPK. This part would be activated by the $Ca^{2+}$ transient which is modelled here as a direct edge from $G_{q/11}$. The third module is in the centre of the network and revolves around the PI3K-PDK1-AKT axis. Most probably activated by free $\beta\gamma$ dimers, it involves downstream activation of p70S6K, CKII and inhibition of CDKs. CKII, AKT and CDK1 account for a significant part of target site phosphorylation. The fourth and fifth modules are the well-known GPCR-activated kinases PKC and PKA. The PKC module has a rather simple structure. PKC is activated downstream of $G_{q/11}$ and activates a sizeable number of targets, but there are very few intermediated kinases leading deeper into the network. The PKA module

is similar, but differs from the PKC module by the presence of downstream intermediate kinases (EEF-2K, STK38, CKI-α, CDK5, STK11 and MARK1) which add more layers and lead to crosstalk with other modules. A comprehensive table with all K-S relationships and PPIs in the networks and PMID numbers for the primary evidence for these edges is provided in Table EV5.

Although only one-third of EDN target sites were shared between the two cell lines tested, the network models for UACC257 (Fig 5) and A2058 (Fig EV3) shared a strikingly similar modular topology. This is in line with the strong overlap of activated kinases (Table 2). Thirty-four intermediary proteins were

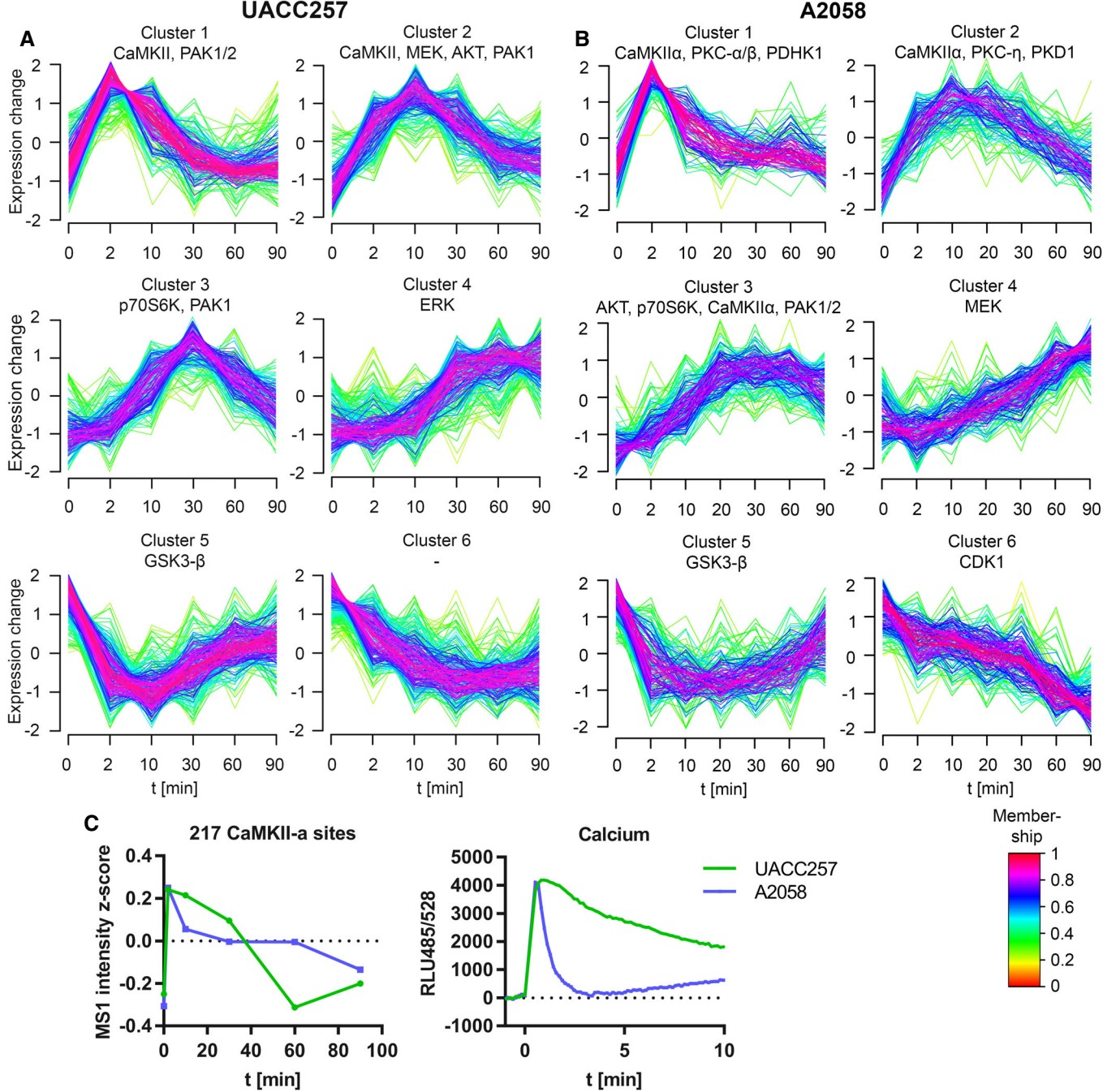

**Figure 4. Target phosphopeptides show six distinct response kinetics, which are shared between cell lines and are associates with similar sets of kinases.**

A, B   EDN target phosphopeptides for (A) UACC257 and (B) A2058 were segregated into six clusters each, using c-means clustering (MFuzz). Each line represents one phosphopeptide, and cluster membership is indicated by colour. Most likely kinases for each phosphosite were predicted using NetworKIN, and enrichment of kinase substrates was calculated for each cluster. Enriched kinases are indicated above each cluster. Note that substrates of these kinases only account for a fraction of all phosphosites in a cluster. Detailed information on the kinase enrichment is provided in Appendix Table S1.

C   Average MS1 intensity *z*-score kinetics for 217 predicted CaMKII-α sites that are shared between both cell lines. For comparison, average (*n* = 6) calcium abundance kinetics from Fig 1A are shown.

shared between the two cell lines, while 22 and 2 proteins were unique to the UACC257 and A2058 models, respectively. The A2058 intermediary nodes constituted a subset of the UACC257

intermediary nodes. On the target site level, the overlap was much smaller. Fifty-two target sites were shared between models, 77 occurred only in the UACC257 model, and 37 target sites were

only found in the A2058 model. However, it cannot be excluded that less well-studied pathways are different between the two cell lines, since kinase activation prediction and network modelling rely on publically available prior knowledge which is biased in favour of well-studied signalling pathways.

To explore whether the similar network structures resulted from constraints imposed by the modelling approach, a control network was generated based on randomisation of the UACC257 phospho-proteomic data set. Quantitative data of the identified EDN target sites were assigned phosphosite identifiers from non-regulated phosphosites. The resulting control network (Appendix Fig S1) did not contain most of the network structure of the UACC257 and A2058 models. However, it did feature PKC, PKA and AKT. In contrast to the UACC257 and A2058 networks, these kinases appeared as connecting nodes rather than central hubs with many substrates. The control network also contained a part of module 3 based on CKII, CDK1 and CDK2. Occurrence of these kinases is probably due to higher frequencies of their substrates in the prior knowledge database. This analysis showed that the network structure was strongly dependent upon the selection of EDN-regulated sites as input for PHONEMeS.

While the modelling approach did not directly build on kinase activation predictions, the same kinases appeared again in the model as central nodes organising the network. For EDNRB signalling, kinase activation prediction and network modelling converged on 18 kinases, suggesting their central role in EDNRB signalling.

## Validating the activation of central kinases of the EDNRB signalling model

Experimental validation of the derived network was performed by analysing the activation of central kinases of the network in response to EDN. For this purpose, phosphorylation of established marker substrates for 12 of the identified central kinases following EDN stimulation was assayed by Western blot, as a proxy for kinase activation.

In addition to the already validated AKT (Fig 1B), the activation of 11 kinases through EDN was tested (Fig 6A–K). Increasing concentrations of specific inhibitors for the kinase under study were used to support the specificity of substrate phosphorylations (Fig EV4A–L). EDN induced phosphorylation of all substrates tested (Fig 6A–K), except for Rb, a model substrate of CDKs, for which PHOXTRACK and NetworKIN predicted inhibition.

Eight of the marker substrates for the activated kinase predictions were not measured in the phosphoproteomic data sets (Rb S807/811, CaMKII S286, MAPKAPK2 T222, cJun S73, cRaf S338, ACC S79, PKA motif and PKC motif). PKA activity and PKC activity were evaluated with motif-specific antibodies recognising a pattern of substrates. Activation of PKA through EDNRB was a surprising finding, since ENDRB activates G$_i$ (Takagi et al, 1995) in the few systems studied until now. The validity of this finding is supported through inhibition of EDN-mediated PKA target motif phosphorylation by the adenylate cyclase inhibitor ddcAMP (Fig EV4A).

In summary, the presented data validate the activation of central nodes predicted by the network model.

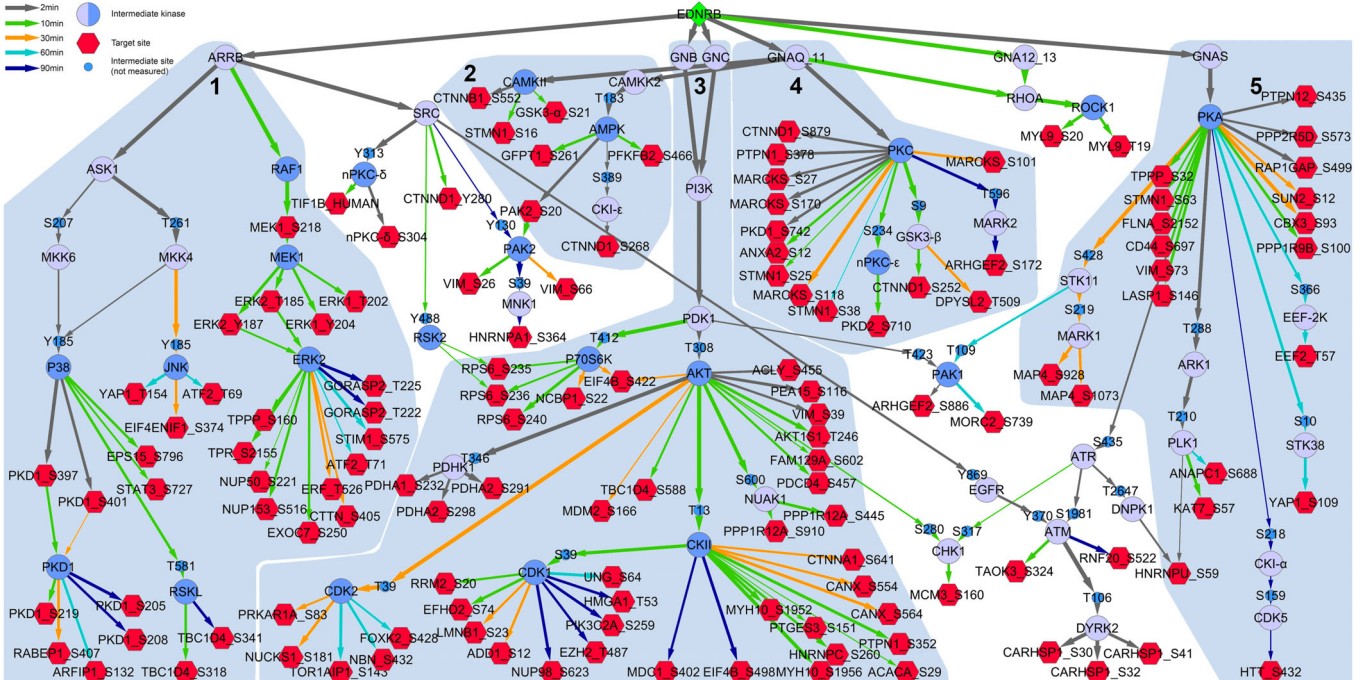

**Figure 5. Prior knowledge-based time-resolved network model of EDN signalling in UACC257 cells.**

Construction of the network is described in detail in Materials and Methods. EDNRB (green diamond) was connected to its target phosphorylation sites (red hexagons) through intermediary kinases or G proteins (blue circles) in a time-resolved variant of the PHONEMeS approach. Central kinases, which were also identified by kinase activation prediction, are shown as intermediary kinases with dark blue shading. Edge thickness corresponds to weights, which were assigned by downsampling the network 100 times. Entry time point was defined as the point at which edge weight reached 20 and is shown as edge colour. The network was divided into five modules, indicated as light blue outlines and labelled 1–5. Common names are shown for kinases, and primary gene names are shown for all other proteins.

In a second approach, parts of the network structure were validated using inhibitors against three of the central kinases—MEK, PKC and p70S6K. The EDN stimulation experiment was repeated for UACC257 cells at a single 20-min time point, and the impact of kinase inhibition on EDN-induced substrate phosphorylation in the network was tested using a separate phosphoproteomic experiment.

Of the 130 phosphosites in the network model, 103 could be detected in the validation experiment. Seventy-two of the detected sites were again found to be altered upon EDN treatment at the single time point investigated. The effects of kinase inhibition are presented as three perturbed EDN signalling networks (Appendix Fig S2).

The three inhibitors blocked activation of almost all direct and downstream target sites for the inhibited EDN target kinases (9/10 for PKC and MEK/ERK, 4/5 for p70S6K) predicted in the model (Appendix Fig S2).

While the p70S6K inhibitor response was constrained exclusively to the predicted targets, additional responses were observed upon inhibitor treatment with the MEK/ERK and PKC inhibitors. This included activating inputs from PKC and MEK/ERK to p70S6K, which could not have been predicted since they were not contained in the prior knowledge. In addition, blocking PKC and MEK/ERK prevented activation of 11 and seven sites, respectively, scattered across the entire network with no discernible pattern.

Mechanistic implications of these experiments should be interpreted cautiously, considering that inhibition of a substrate phosphorylation may be the result of (i) crossreactivity of the kinase inhibitor with unrelated kinases or (ii) indirect effects between inhibited kinase and substrate through intermediary kinases not included in the model or (iii) adaptive responses of inhibitor treatment (e.g. prevention of constitutive phosphorylations of unrelated

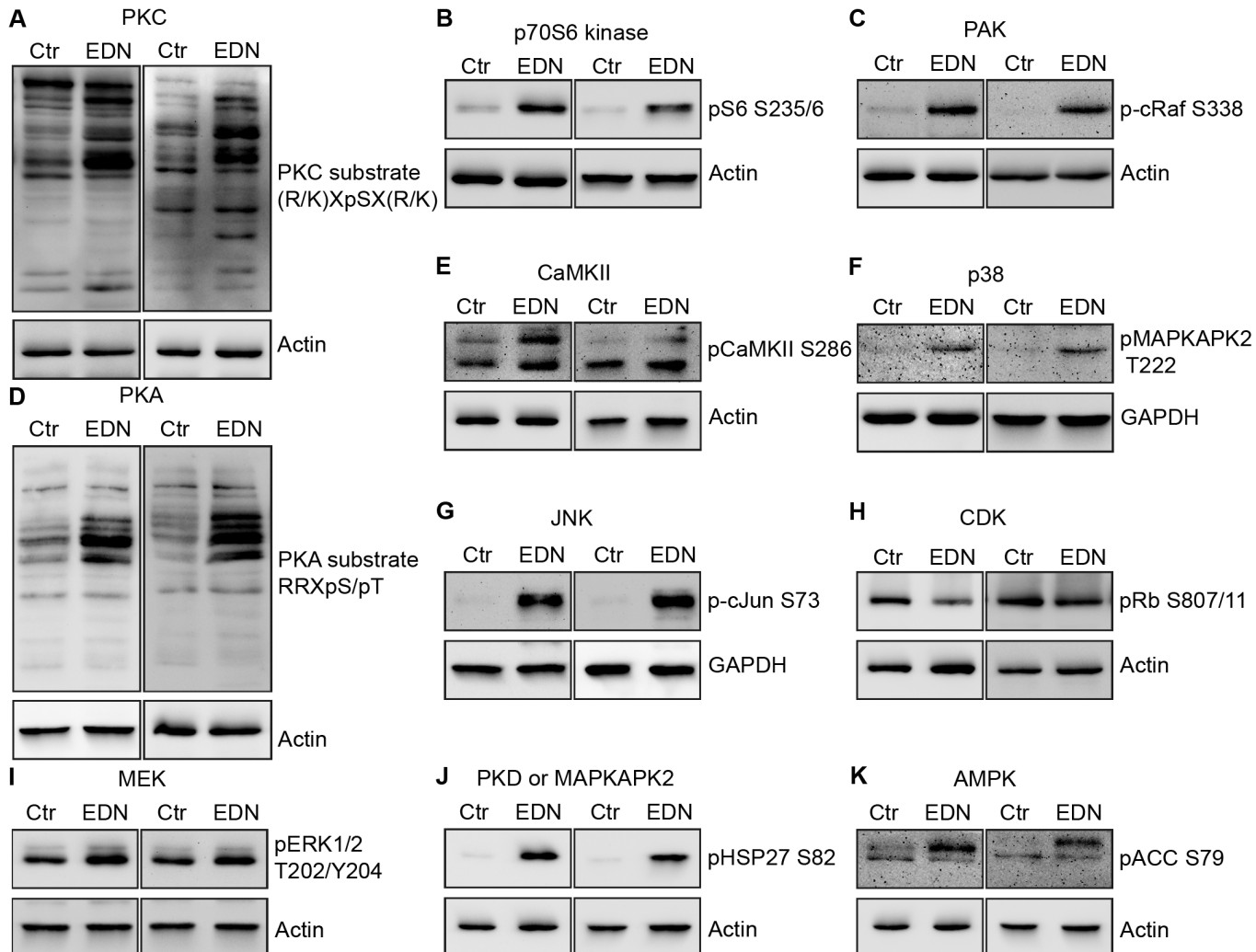

**Figure 6. Validation of kinase predictions using marker substrates.**

A–K   Predicted kinases are indicated on top of each panel for which one of their marker substrate phosphorylations was assayed. UACC257 cells were serum starved and stimulated with PBS or EDN at 100 nM for 60 min (5 min for CaMKII). Ten micrograms protein from RIPA lysates was analysed by Western blot. Kinase marker substrate phosphorylations and corresponding loading controls are shown. For each kinase, two independent replicates were performed and are shown. Each gel also had control lanes with increasing concentration of kinase inhibitors to support phospho band identity (extended versions with inhibitors are shown in Fig EV4).

kinases). Notwithstanding these known limitations, the results from the inhibitor experiments validate the network structure for the three tested kinases and their importance for EDN signalling predicted by the model.

### EDN-induced melanoma cell migration depends on activation of AKT, JNK, PKC and AMPK

Finally, systemic insights from the EDNRB signalling model were used to alter relevant cellular functions controlled by EDNRB signalling. Migration and reorganisation of the cytoskeleton were the most prominent hits in the pathway enrichment analysis (Fig 3) and play an important role for EDN in melanoma. Pharmacological inhibition of kinases identified by the model as central nodes of the EDNRB network was used to test their role in EDN-induced melanoma cell migration.

Using specific inhibitors for 17 of the 18 central kinases (Appendix Table S2; RAF, MEK and ERK were among the 18 central kinases, but ERK was not directly targeted), different branches within the five modules of the signalling network were blocked. Titration of the inhibitors established their efficacy for blocking substrate phosphorylation at 0.1 μM or 1 μM (Fig EV4A–L). Migration was then measured with a wound healing assay and automated microscopy (Fig 7A). Inhibition of the four kinases PKC, AKT, AMPK and JNK blocked EDN-induced cell migration (Fig 7B), while inhibition of all other kinases tested could not prevent this effect. The four inhibitors only blocked EDN-induced cell migration, without influencing basal migration (Fig 7B). A summary of the result for 16 inhibitors is given as a heatmap (Fig EV5A). The CDK inhibitor was excluded because it caused complete cell death after 24 h (Fig EV5C). Among the other inhibitors, only BRAF and MEK inhibitors reduced cell viability (Fig EV5C). The presented data show migration after 48 h. To exclude that closure of the scratch was caused by proliferation rather than migration, proliferation was measured with a BrdU assay. EDN did not increase proliferation for up to 72-h treatment (Fig EV5B). When related back to the network model (Fig 5), it becomes obvious that the four kinases controlling cell migration are in modules 1–4 of the network. This implies that control of EDN-induced migration involves a concerted signalling process involving almost the entire network, rather than a single sequential pathway. In contrast, the 12 kinases which could not be linked to EDN controlled cell migration were in parts of parallel branches of modules 1 (RAF, MEK, p38, PKD, RSK) and 2 (CaMKII) as well as downstream kinases in module 3 (CKII, p70S6K, PAK1), PKA in module 5, ROCK and MAPKAPK2.

EDN-induced migration has been linked to changes in adhesion molecule expression, specifically downregulation of E-cadherin (Bagnato *et al*, 2004). Measurement of E-cadherin with an SRM assay showed that only the AMPK inhibitor, out of 17 kinase inhibitors tested, could significantly inhibit E-cadherin repression by EDN (Fig 7C).

In conclusion, the signalling model proved a successful tool to identify kinases through which the EDNRB signalling network controls melanoma cell migration. Integrative modelling of phosphoproteomic changes enabled these key findings, which would not have been accessible by protein or RNA expression profiling, given the absence of measurable protein expression changes. These results show that insights from the EDNRB signalling model can be used to alter phenotypic properties of melanoma cells which are controlled by EDN signalling.

## Discussion

The first unbiased quantitative phosphoproteomic study of the pharmacologically relevant EDN signalling pathway is presented. EDNRB activation elicited a characteristic pattern of phosphorylation changes affecting, among others, distinct functional protein classes controlling cytoskeleton organisation, cell signalling and cell motility. The high quality of the obtained data allowed for the prediction of a specific set of target kinases activated downstream of EDNRB. Utilising the comprehensive time-resolved phosphorylation analysis, we established a logic model of the EDN signalling network comprised of five modules. Several key kinases predicted by the model were biochemically validated and functionally linked to EDN-induced cell migration.

Analysis of EDNRB signalling and the logic model were based on two phosphoproteomics data sets with high temporal resolution. These data sets contain accurate quantification covering over 5,000 phosphorylation events measured across five time points in biological triplicates. Comparison of observed phosphorylation changes to mock-stimulated controls as well as EDNRB knockout controls ensured confident identification of EDN target sites and clearly demonstrated that all observed changes occurred downstream of EDNRB. The obtained time-resolved phosphorylation data following activation of a single GPCR type provided the ideal basis to develop the first EDNRB logic signalling model.

Logic modelling of quantitative phosphorylation data is an emerging strategy to better understand the complex nature of cellular signalling systems and has the potential to improve research on the control of pleiotropic signalling responses. EDN controlled cellular reactions are often context dependent, and EDN signalling has to be integrated with concurrent cues, including hormones, paracrine factors, cell–cell contracts and cell-matrix contacts. Reductionist signalling models based on the sequential analysis of signalling components using classical biochemical assays are not well suited to reflect the complex signal integration through which cellular reactions are determined. Integrative approaches based on global data collection and curated prior knowledge are a better strategy to more accurately model the complex organisation of cellular signalling systems to enhance predictive knowledge on the context dependency and diversity of functional outcomes. In this study, integrative modelling was applied for the first time to the EDN signalling system.

Despite the coverage of hundreds of EDN-induced changes in protein phosphorylation, the presented data sets are incomplete as the phosphoproteome is much more extensive. Although the prior knowledge background covered over 15,000 reactions, it was limited to well-established signalling processes. In spite of these limitations, we are confident that our approach captured the core modules of EDNRB signalling because (i) the core model components and their interactions are in general agreement with previous studies using classical approaches. (ii) Predictions based on the model could be functionally validated. (iii) The network models resulting from the two analysed cell lines were strikingly similar despite the moderate overlap in

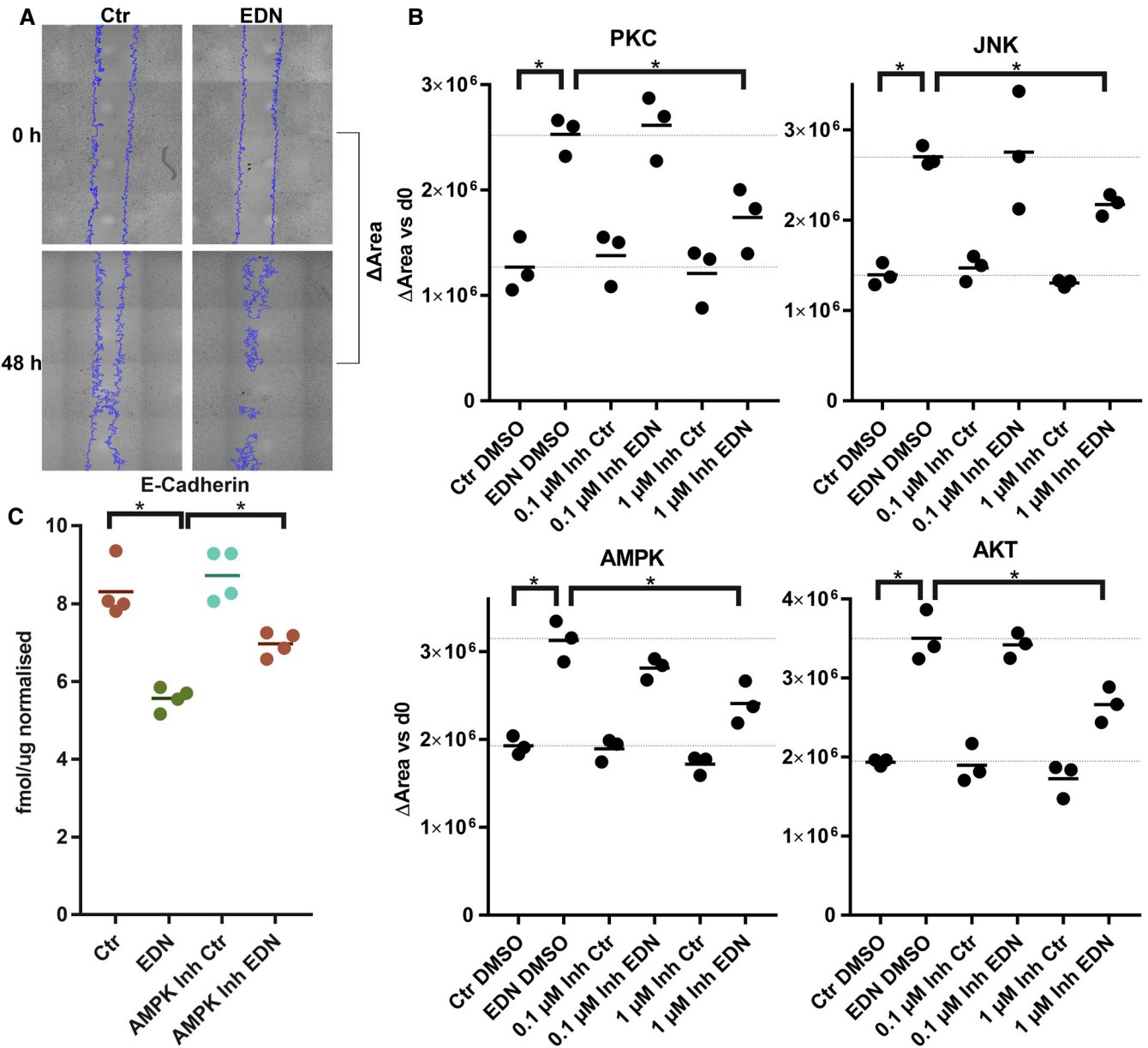

**Figure 7.  PKC, JNK, AMPK and AKT mediate EDN-induced cell migration.**

The effect of kinase inhibitors on EDN-induced cell migration of UACC257 cells was analysed using a scratch assay. Scratches were made in UACC257 monolayers. Serum was removed, and cells were treated with PBS or 100 nM EDN in combination with DMSO or kinase inhibitors at 0.1 µM or 1 µM in triplicates.

A   Example pictures showing the effect of EDN on migration. Cell free areas are delimited by blue lines. Migration was quantified as the change in cell free area between 0 and 48 h.
B   The four inhibitors that were effective at 1 µM are shown in detail, i.e. every point corresponds to one well.
C   Abundance of E-cadherin (normalised to GAPDH and Actin) was assessed by LC-SRM after 48-h treatment with EDN or kinase inhibitors at 1 µM. Only the single inhibitor which could reverse EDN-induced E-cadherin repression is shown. The full SRM data set is provided in Appendix Fig S3.

Data information: *$P < 0.05$ two-sided unpaired $t$-test.

phosphoproteomics data. These results are in line with a meta-analysis of publicly available phosphoproteomics data set, showing that the same cell perturbations in different studies led to the same kinase activation predictions, although the actual phosphopeptides underlying these predictions were different (Ochoa et al, 2016).

The EDNRB signalling model is more expansive and coherent than the current heterogeneously assembled models of EDN signalling (Bouallegue et al, 2007; Rosano et al, 2013). Moreover, it is in accordance with a number of known mechanisms activated downstream of EDNRA and EDNRB supporting the overall robustness and reliability of the study. Activation of EDN receptors is known to

cause $Ca^{2+}$ release and activate PKC through $G_q$ as well as influencing cAMP levels and modulating PKA activity (Rosano *et al*, 2013). All three of these mechanisms are contained in the network modules 4 and 5. PI3K-AKT activation is well established for ENDRA (Bouallegue *et al*, 2007). Although EDNRB activation of PI3K-AKT has not been as extensively studied, it was shown to involve free $G_{\beta\gamma}$ dimers in endothelial cells (Liu *et al*, 2003), which is in accordance with module 3 of the network model. The calcium-dependent kinase module 2 has not been well studied for EDN signalling; it contains two branches, the first consisting of CaMKII and the second of CaMKK2 and AMPK. CaMKII activation by $Ca^{2+}$ release downstream of EDNRB is supported by the correlation of the calcium transient with CaMKII substrate phosphorylation. While AMPK is not commonly associated with calcium signalling, it has recently been recognised as part of the CaMKII independent, calcium-dependent pathway downstream of CaMKK2 predicted by the model (Frigo *et al*, 2011).

Module 1 encompasses activation of the three MAPK cascades leading to ERK, JNK and p38 activation, which was experimentally validated here and is in agreement with previous studies. While the mechanism through which GPCRs activate the MAPK cascades is incompletely understood (Marinissen & Gutkind, 2001), the modelling algorithm predicted a central role for arrestin. There are three alternative mechanisms for MAPK activation by EDN, which differ in how the EDN receptor activates RAF. The first mechanism is based on arrestin acting as a scaffold bringing RAF and ASK (MP3K5) in close proximity to their downstream kinases, thus enhancing their sequential phosphorylation (DeWire *et al*, 2007). The second mechanism involves activation of RAF by PKC (Smith *et al*, 2017). The third mechanism depends on transactivation of EGFR by arrestin-mediated src activation (Rosano *et al*, 2013). Only the first mechanism is selected by PHONEMeS, but the other two mechanisms are valid possibilities. This mechanism gets selected because arrestin leads to activation of three MAPK cascades from a receptor proximal node. The second mechanism involves a much longer path to RAF and would not explain JNK and p38 activation. The third mechanism starts with arrestin activation which then has a shorter path to RAF and ASK without involving src and EGFR. Although arrestin activation downstream of EDNRA has been shown (Rosano *et al*, 2009), the second and third mechanisms are favoured in the EDN field. Nevertheless, the three mechanisms only differ in how the MAPK module is connected to EDNRB and are thus all compatible with the overall structure of the network model.

In contrast to lists of regulated phosphosites, the EDNRB signalling model better represents the complex nature of the interplay of concurrently affected branches which orchestrate downstream processes. The present study substantially expands the knowledge about EDNRB signalling, uncovering a previously unknown spectrum of target kinases and target phosphosites, while at the same time proposing an organising structure explaining these changes. As such, it can serve as a blueprint to guide future studies on EDNRB signalling. This opens many new possibilities in the EDN field: studies into biological functions controlled by EDN can use the model as an overview which pathways are engaged by the receptor. Mechanistic studies can extend the proposed structure with new insights into signalling mechanisms. Studies investigating crosstalk between EDN and other signalling molecules may overlay the model with different pathway maps to identify likely points of crosstalk.

Though phosphoproteomic methodology has matured substantially in the last decade, generation of meaningful signalling networks from these studies is just becoming possible due to computational advances. Manual integration of hundreds of phosphorylation events with thousands of prior knowledge reactions is obviously not feasible. The advances that led to the model presented here were organisation and formalisation of the accumulated knowledge on signalling pathways from thousands of individual publications into databases (Turei *et al*, 2016) and the development of new algorithms for network interference (Terfve *et al*, 2015). The presented workflow is readily applicable to other medically relevant GPCR signalling systems and will be of broader utility to generate network models in different contexts.

The insights provided by the EDN signalling model were used to enable targeted disruption of the signalling pathway to better understand the mechanism of EDN-induced cell migration and how to control it. This basic process is of crucial importance during invasion and metastasis formation. Recent studies have confirmed that EDNRB is a driver of metastasis in a melanoma mouse model (Cruz-Munoz *et al*, 2012), a property originating from the ability of EDN to induce cell migration (Scott *et al*, 1997). In addition to its autocrine effect, melanoma cell-derived EDN can activate EDNRB on endothelial cells to induce angiogenesis and neovascularisation in a different mouse model (Spinella *et al*, 2014).

Among the central kinases of the signalling model, AKT, JNK, AMPK and PKC were linked to EDN-induced tumour cell migration. To our knowledge, PKC and AMPK have not been associated with EDN-induced cell migration until now. On the other hand, AKT and JNK were already known to take part in the regulation of this EDN phenotype. Both AKT and JNK inhibition blocked EDN-induced glioma cell migration (Hsieh *et al*, 2014), and AKT inhibition prevented EDNRA-mediated migration of hepatocellular carcinoma cells (Cong *et al*, 2016).

The four kinases identified could be candidate drug targets for combination therapy with EDNRB inhibitors to treat melanoma progression by reducing melanoma cell spreading and metastatic potential. This provides one example how the developed network model can be used to assist in rational, informed drug combination selection. Other interesting applications to be explored are interactions or synergies between EDN receptor blockers and drugs targeting branches contained in the EDN signalling model to modulate other phenotypic outcomes controlled by EDN.

Considering the established impact of EDN signalling on many different physiological and pathological processes, the presented data and model contain great potential for the future development of novel biomedical applications of EDN signalling.

# Materials and Methods

### Material

Chemicals were purchased from Sigma-Aldrich unless otherwise stated. Cell culture supplies were purchased from Life Technologies. Endothelin 1 (CSCSSLMDKECVYFCHLDIIW) was purchased from

Bachem (Product No. 4040254) and is referred to as EDN for the purpose of all stimulation experiments. The following key reagents were employed in the study: TiO$_2$ (GL Sciences), $^{13}C_6$-$^{15}N_2$ lysine (Lys-8, Silantes GmbH) and $^{13}C_6$-$^{15}N_4$ arginine (Arg-10, Silantes GmbH). Heavy peptides for SRM were synthesised by JPT Peptide Technologies. Kinase inhibitors were purchased from Selleckchem, except for ddcAMP (Enzo Life Sciences), GSK-429286, CRT0066101 and JNK-IN-8 (Sigma-Aldrich). Primary antibodies and anti-rabbit secondary antibodies (7074) were purchased from Cell Signalling Technologies [pAKT S473 (9271), pCREB S133 (9198), PKC motif (6967), PKA motif (9624), pS6 S235/6 (4858), pRB S807/11 (8516), CaMKII S286 (12716), pMAPKAPK2 T222 (3316), p-cJun S73 (3270), p-cRAF S338 (9427), pACC S79 (11818), pHSP27 S82 (9709), AKT (4691) and CREB (9197)]. Antibodies against β-Actin (AC-15), GAPDH (MAB374) and EDNRB (21,196) were purchased from Sigma-Aldrich, Millipore and Santa Cruz, respectively. Anti-mouse and anti-goat secondary antibodies were purchased from Jackson ImmunoResearch.

## Cell culture and cellular assays

UACC257 and A2058 cells were obtained from NCI and ATCC, respectively. Cells were cultured in DMEM 10% FCS and routinely checked for mycoplasma. Cell line identity was confirmed by STR sequencing. A2058 EDNRB knockout cells (ENDRB-KO) were generated using the TALEN technology (Cellectis). UACC257 ENDRB-KO cells were generated using CRISPR/Cas9. UACC257 cells were transfected with three plasmids encoding CAS9, a target-specific 20 nt guide RNA and GFP (Santa Cruz) using Lipofectamine 3000 (Thermo Fisher Scientific). Single clones were obtained by limiting dilution. Gene disruption was verified by gene sequencing and Western blot for UACC257 and A2058 (Fig EV1A).

Cells were SILAC labelled in heavy medium (Lys-8 and Arg-10) for five passages, and complete incorporation of isotopes was validated by LC-MS/MS. Cells grown in heavy medium were used to generate the global internal standard (GIST) which was spiked into phosphoproteomic samples for normalisation. Calcium release and cell proliferation were measured using the Fluo-4 NW Calcium Assay Kit (Thermo) and BrdU ELISA (Roche), respectively. Fluo-4 fluorescence (Calcium kit) and TMP staining (BrdU kit) were measured with a Synergy HT plate reader (BioTek).

## EDN stimulation of melanoma cells

For phosphoproteomic and SWATH analysis, $2.5 \times 10^6$ cells were seeded on 14-cm dishes. After allowing the cells to adhere for 24 h, they were serum starved overnight and stimulated with DMEM with or without 100 nM EDN for 2–90 min. Cells were scraped into 1 ml lysis buffer (8 M urea, 100 mM Tris, pH 8, Sigma phosphatase inhibitors 2 & 3), DNA was sheared using an ultrasonic rod, and protein content quantified by BCA assay (Thermo). For phosphoanalysis, 700 μg light lysates were mixed with 400 μg GIST. No GIST was added for SWATH protein expression analysis. Samples were precipitated with methanol/chloroform, resuspended in lysis buffer and subjected to tryptic digestion.

For kinase inhibitor network validation experiments, $2.5 \times 10^6$ UACC257 cells were seeded on 14-cm dishes and serum starved overnight. Cells were treated with DMSO, 100 nM trametinib, 1 μM

Go 6983 or 1 μM LY2584702 for 1 h. Final DMSO concentration for all plates was 0.1%. Subsequently, cells were mock-stimulated or treated with 100 nM EDN for 20 min. All conditions were performed in triplicate. Lysis and further processing were performed as described for phosphoproteomic analysis.

For SRM analysis, UACC257 cells were seeded onto 6-well plates at 180,000 cells per well. The next day, serum was removed from the medium and cells were treated with EDN or vehicle in combination with kinase inhibitors or DMSO (final concentration 0.1%) for 48 h in triplicates. Proteins were extracted using 8 M urea in 100 mM Tris pH 8 and subjected to tryptic digestion.

## Tryptic digestion and phosphopeptide enrichment

Ten micrograms (SWATH, SRM) or 600 μg protein (Phosphoproteomics) was reduced with dithiothreitol (DTT) at a final concentration of 5 mM for 30 min at 37°C and alkylated using iodoacetamide at a final concentration of 10 mM for 30 min. Alkylation was stopped by increasing the DTT concentration to 10 mM. Urea was diluted to 6 M with 100 mM Tris pH 8, and samples were digested with Lys-C at 1/100 enzyme/protein for 2 h at 37°C. Afterwards, samples were diluted to reduce urea below 2 M and digested with trypsin at enzyme/protein 1:100 for 12–16 h at 37°C. Digestions were quenched with 0.5% trifluoroacetic acid (TFA).

Phosphopeptide enrichment was performed as described in Zhou *et al* (2013). Peptide digests were desalted on a 100 mg SepPak C18 cartridge according to manufacturer's instructions. After speedvac drying, samples were reconstituted in 75 μl of loading buffer (80% acetonitrile (ACN), 6% TFA). TiO$_2$ microcolumns were prepared from constricted GELoader tips and packed with 200 μg TiO$_2$ material. Samples were loaded onto the microcolumns for ~ 30 min at 100 g. After washing with 100 μl loading buffer and 100 μl 50% ACN 0.1% TFA, phosphopeptides were eluted with 30 μl 5% NH$_4$OH followed by 5 μl 50% ACN, 0.5% acetic acid. Finally, samples were cleaned-up using C18 MicroColumns (NestGroup) according to manufacturer's instructions.

## Data-dependent acquisition (DDA) phosphopeptide analysis

DDA phosphopeptide data were acquired on a LC-MS/MS system consisting of a Proxeon Ultra easy LC and an Orbitrap Elite (Thermo). Peptides were separated on a PepMap100 column (C18, 0.075 × 150 mm, 2 μm, 100 A) with solvent A: 5% ACN, 0.1% formic acid (FA) in water and solvent B: 98% ACN, 0.1% FA. Gradient settings were 0–120 min: 5%B–25%B at 300 nl/min. The Orbitrap Elite was run in data-dependent mode with parallel MS1/MS2 acquisition. Survey full scan MS1 spectra (from *m/z* 350 to 1,600) were acquired in the Orbitrap with resolution *R* = 120,000 at *m/z* 400. Up to 15 ions with charge state ≥ +2 were selected for fragmentation per cycle with a dynamic exclusion window of 30 s.

## Data analysis to generate phosphoproteomics data sets

Peptides were identified using the Trans-Proteomic Pipeline (TPP) v4.7 with search engines Comet, OMSSA, MyriMatch and XTandem with the parameters: precursor tolerance: 10 ppm; fragment tolerance: 0.5 Da; static modifications: iodoacetamide (C); dynamic

modifications: Label:13C(6)15N(2) (K); Label:13C(6)15N(4) (R), phospho (STY), oxidation (M); enzyme: Trypsin; missed cleavages: 2. Posterior probabilities were assigned with PeptideProphet and iProphet, and a peptide FDR filter of 1% was applied. MS1 XIC quantification of light and heavy peptide pairs was performed in Skyline v3.6 using the MS1 filtering workflow. In parallel, phosphate localisation probabilities and corrected sequence assignments were calculated using LuciPHOr2 (Fermin *et al*, 2013). Quantification and localisation results were merged in Excel 2013, and data sets were filtered to false localisation rate (FLR) 1%. Enrichment specificity values scattered around 85% phosphopeptide identifications (Fig EV1B), as expected for $TiO_2$ enrichment.

The presented data sets contain two groups of phosphopeptides: The first consists of peptides with phosphates localised at high confidence, and the second consists of peptide groups with a certain backbone sequence and number of phosphates but without high confidence phosphate site localisation. The latter are often removed from phosphoproteomics data sets when a localisation cut-off is applied. In this paper, these peptides are retained as a separate category, because they can still indicate regulatory processes. The characteristics of the LuciPHOr results are shown in Fig EV1C. The difference between the curve for peptides with localised phosphates and for groups of peptides with the same sequence and number of phosphates (delocalised) gives the number of positional (isobaric) isomers for a certain FLR threshold. These curves converged at FLR 1%, indicating that most of the phosphopeptides have only one high confidence localisation pattern. At an FLR threshold of 1%, about 20% of observed phosphopeptides for which no localisation can be determined with high confidence would be discarded (difference between the horizontal dashed lines). These phosphopeptide groups were retained, and the most likely phosphate positions assigned by LuciPHOr were used for all subsequent analyses.

Normalisation was performed in two steps using the GIST. First, errors in mixing light samples and GIST were compensated by median normalisation of the light-to-heavy (L/H) ratio distributions over all samples. Second, peptide-specific normalisation was performed for every light peptide MS1 intensity using the corresponding heavy peptide intensities. The reproducibility of quantification was benchmarked by comparing the coefficient of variation (CV) for six process replicates, starting from one lysate (Fig EV1D). Light phosphopeptide MS1 intensity CVs before (LFQ) and after normalisation to the GIST (SILAC GIST) showed an improvement from 29.3 to 12.3% median CV (Fig EV1D). Control and EDN-stimulated groups were compared using two-tailed unpaired *t*-tests of normalised peptide intensities for every time point, and fold changes were calculated. *P*-values were corrected for multiple testing using Benjamini–Hochberg multiple testing correction.

Unless otherwise stated, phosphopeptide data sets (Tables EV1 and EV2) were used for analyses. These tables contain peptides with confidently (FLR < 1%) and ambiguously (FLR > 1%) assigned phosphate localisations, respectively. For the latter, all positional isomers were grouped and only the most likely localisation was used. For kinase activation prediction and network modelling, phosphopeptide tables were converted to phosphosite tables (Tables EV1 and EV2) assigning one row to each site in the data set.

## DIA-SWATH proteome quantification

SWATH analysis was performed on a TripleTOF 6600 (AB Sciex) coupled to an Ekspert nanoLC 400 autosampler (Eksigent) and a 1D+ nanoLC-ultra pump (Eksigent). Samples were spiked with iRT peptides (Biognosys), and peptides were separated on a 40-cm self-packed emitter (0.075 μm inner diameter PicoFrit, New Objective) packed with C18 ProntoSIL 200 3 μM AQ, 200A) using a linear 60-min gradient from 5 to 35% buffer B (98% ACN, 0.1% FA) in buffer A (2% ACN, 0.1% FA). The TripleTOF 6600 was operated in SWATH mode. Each cycle consisted of a 200 ms MS1 scan and 64 variable window MS2 scans, spanning the precursor mass range between 400 and 1,200 *m/z* with 50 ms per scan, yielding a cycle time of 3.4 s.

Peptide peak groups were extracted and scored from the SWATH runs using OpenSWATH (Rost *et al*, 2014) against a pan-human spectral library (Rosenberger *et al*, 2014) with RT extraction window 600 s, *m/z* extraction window 0.05 Th and iRT recalibration enabled. The data set was then filtered with SWATH2Stats v1.6.1 (Blattmann *et al*, 2016) to obtain a protein FDR of 2% using a decoy counting approach. Finally, a median normalised protein MS1 intensity matrix was generated using mapDIA v2.4.2. Differential expression was tested by two-tailed unpaired *t*-tests of MS1 intensities followed by multiple testing correcting (Benjamini–Hochberg).

## LC-SRM analysis

Following digestion, Lys-8/Arg-10 labelled standard peptides for E-cadherin, β-actin and GAPDH were added at defined concentrations before C18 MicroColumn clean-up. SRM analysis was performed on a TSQ Vantage (Thermo) coupled to a nanoLC Ultra1D+ (Eksigent). Peptides were separated on a column (0.075 × 100 mm, C18 ProntoSIL 200 3 μM, 200A) packed into a PicoTip Emitter (New Objective). Peptides were separated along a linear gradient of B (98% ACN, 0.1% FA) in A (2% ACN, 0.1% FA), running from 2 to 45% B in 40 min. The TSQ was run in scheduled SRM mode, detecting three peptides per protein with three transitions for the light and heavy forms. Target peptides were quantified by their L/H XICs in Skyline v.4.1. E-cadherin was normalised to the average abundance of beta-actin and GAPDH and exported to GraphPad Prism v7.03.

## Pathway enrichment and kinase predictions

A core analysis of each phosphoproteomics data set was performed in Ingenuity Pathway Analysis (IPA; Build 470319M) using "phosphorylation analysis" in the advanced analytics module. More than 98% of phosphosites in each data set mapped to a protein in IPA. For all analyses, the experimentally observed phosphopeptides were used as the reference set. The differential abundance *q*-value cut-off applied was 0.1. Result tables for the categories "Canonical Pathways" and "Disease and Biofunctions" were exported for each time point and merged. A filter requiring two time points with significant enrichment ($P < 10^{-3}$) was applied, and the resulting table was plotted in GraphPad Prism v7.03.

PHOXTRACK analysis was performed on phosphosite data sets (Tables EV1 and EV2, see Data analysis to generate phosphoproteomics data sets) separately for each time point against all available databases with 10,000 permutations and minimum three substrates per kinase. For NetworKIN-based predictions, abundances of target sites were averaged between replicates. The resulting curves were split into six clusters using c-means clustering implemented in the Mfuzz package v2.4 in R v3.4. Most likely kinases were predicted using NetworKIN 3.0 (Linding *et al*, 2008). Kinase–substrate enrichment in each cluster over all identified sites in the data set was calculated in R v3.4, using Fisher's exact test followed by Benjamini–Hochberg multiple testing correction.

### Signalling network modelling

EDNRB signalling networks were generated with a modified version of PHONEMeS (Terfve *et al*, 2015) using phosphosite data sets (Tables EV1 and EV2) as input. In its previous implementation, PHONEMeS (https://saezlab.github.io/PHONEMeS/) was used to build and train Boolean logic models of signalling networks downstream of a perturbed kinase by combining phosphoproteomics data set with a space of K-S relationships. Differentially abundant phosphopeptides were identified by statistical testing with multiple testing correction at $qThresh = 0.1$ and FC > 1.5 up or down. To each of the measurements $i$ for each of the time points $j$, a score was assigned based on $q$-value $s_{i,j} = \log_2(q_{i,j}/qThresh)$. In the present study, PHONEMeS was reformulated as an Integer Linear Programming problem, enhancing model optimisation speed by orders of magnitude (Gjerga *et al*, manuscript in preparation).

With PHONEMeS, paths from EDNRB to perturbed sites were inferred using a Boolean logic modelling scheme. The network was trained by rewarding the inclusion of correctly predicted perturbations and penalising the inclusion of non-regulated sites in the network. The ILP formulation consisted of two main parts: an objective function through which the overall sum of scores assigned to each measurement was minimised and a set of linear constraints. Because of the minimisation procedure, the optimal solution network incorporates phosphosites significantly altered after EDN treatment, while penalising the inclusion of the rest of the measured sites. The set of constraints was used to formulate the rules in which perturbation propagates from EDNRB and how it connects to the downstream sites. To adapt the method to GCPR signalling, PHONEMeS was applied as follows: first, the background K-S network derived from Omnipath (Turei *et al*, 2016) was complemented with a subset of all directed and signed PPIs from Omnipath (Turei *et al*, 2016) associated with GPCR signalling in Reactome (Pathway R-HSA-372790). Closely related heterotrimeric G protein subunits and kinase isoforms were grouped in the prior knowledge database (Table EV5). Second, PHONEMeS was extended to handle time-resolved phosphoproteomics data. The network was generated by running PHONEMeS successively for every time point, building on the network structure of the previous time point. This process was repeated 100 times where on each iteration the data were randomly downsampled with replacement. For each run, this procedure retained 61–65% unique sites that were considered for the analysis. The individual models were then combined, and weights were assigned to each of the interactions based on how often an interaction appeared in the individual models. These multiple runs enabled the capture of multiple alternative network solutions, and the most likely interactions were assigned a higher weight. All network modelling steps were performed in R v3.4 and visualised using Cytoscape v3.3.

### Western blots

Ten micrograms total protein were separated by SDS–PAGE and semidry-blotted onto nitrocellulose membranes. Membranes were blocked in TBS-T 5% skimmed milk for 1 h at room temperature and incubated with primary antibodies diluted in TBS-T 5% skimmed milk overnight at 4°C. Membranes were washed three times in TBS-T and incubated with horseradish peroxidase coupled secondary antibodies in TBS-T 5% BSA for 1 h at room temperature. After washing the membranes four times in TBS-T, ECL was added and pictures were documented with an Alpha Imager (Alpha Innotech).

### Migration assay

UACC257 cells were seeded on 24-well plates pre-coated with gelatine to form a closed monolayer in full medium. The next day, scratches were generated with Wound Healing Assay plastic inserts (Cell Biolabs). Wells were washed with PBS three times to remove serum and scraped cells and treated with DMEM supplemented with EDN and kinase inhibitors at 0.1 μM or 1 μM where appropriate. Final DMSO concentration in all wells was 0.1%. Wells were imaged with a MD2 Image Xpress Microscope (Molecular Devices) with 15 pictures at 4× magnification after 0, 24 and 48 h. At the end of the experiment, cells were stained with MTT to assess cell viability. Images for each well were stitched together in ImageJ v1.5.1, and cell free areas were quantified using the ImageJ macro (http://dev.mri.cnrs.fr/projects/imagej-macros/wiki/Wound_Healing_Tool).

## Data availability

The data sets and computer code produced in this study are available in the following databases: Proteomics data: PRIDE PXD012316 (https://www.ebi.ac.uk/pride/archive/projects/PXD012316). Modelling scripts: Github (https://github.com/saezlab/EDN_phospho). Networks models: Biomodels MODEL1904170001 (https://www.ebi.ac.uk/biomodels/MODEL1904170001).

*Expanded View* for this article is available online.

### Acknowledgements

We thank Ludovic Gillet, Federico Uliana and Alexander Leitner for operation and maintenance of the Orbitrap and 6600 TripleTOF LC-MS systems as well as Benedicte Haenig for the generation of the EDNRB CRISPR/Cas9 knockout cell lines. We would like to acknowledge the ScopeM microscopy facility for technical support and specifically Roger Meier for support with the screening microscope. This study was funded by Idorsia Pharmaceuticals. E.G. was supported by the European Union's Horizon 2020 research and innovation programme (675585 Marie-Curie ITN "SymBioSys"). R.A. acknowledges the following grant support: ERC grant Proteomics 4D (670821), and the Swiss National Science Foundation (3100A0-688107679). The research of M.G. has

been supported by the Innovative Medicines Initiative project ULTRA-DD (FP07/2007-2013, Grant no. 115766).

## Author contributions

PMAG, FL, MG, IR, RWDW and AS designed the study; MG supervised the study; AS and IR performed experiments; IR and FL contributed new reagents or analytic tools; EG and JS-R developed and implemented the modelling approach; AS wrote the paper; RWDW, IR, MG, RA, JS-R and EG provided critical discussion and edited the paper.

## Conflict of interest

Richard W.D. Welford, Imke Renz, Francois Lehembre and Peter M.A. Groenen are employees of Idorsia Pharmaceuticals, a company which develops medicines targeting the endothelin system.

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
