## [Review Process File · Molecular Systems Biology]

Elucidating essential kinases of endothelin signalling by logic modelling of phosphoproteomics data

Alexander Schäfer, Enio Gjerga, Richard W. D. Welford, Imke Renz, Francois Lehembre, Peter M. A. Groenen, Julio Saez-Rodriguez, Ruedi Aebersold and Matthias Gstaiger

Review timeline:	Submission date:	17 January 2019
	Editorial Decision:	5 March 2019
	Revision received:	17 May 2019
	Editorial Decision:	4 July 2019
	Revision received:	9 July 2019
	Accepted:	11 July 2019

Editor: Jingyi Hou

Transaction Report:

1st Editorial Decision

5 March 2019

Thank you for submitting your work to Molecular Systems Biology. We have now heard back from two of the three referees who agreed to evaluate your manuscript. Since their recommendations are quite similar, I prefer to make a decision now rather than further delaying the process. If I receive the comments from referee #1, I will send them to you and you can address the issues raised by referee #1 together with those raised by the other two referees. You will see from the comments below that the referee #2 and #3 find the manuscript to be of interest and are cautiously supportive. They raise, however, several important points, which should be convincingly addressed in a revision of this work.

Without repeating all the points raised in the reviews below, we think the major issues raised are the following:

- Some of the network predictions should be experimentally validated with inhibitor treatment followed by phosphoproteomic analysis as suggested by referee #2.
- More details and clarifications should be provided with regard to how the models are built as referee #3 indicated.

A revised manuscript will be once again subject to review and you probably understand that we can give you no guarantee at this stage that the eventual outcome will be favorable.

 REFERENCE REPORTS

Reviewer #2:

The manuscript presented by Schäfer et al. is an interesting resource paper that provides data describing the phosphoproteomic landscape of melanoma cell lines after stimulation with endothelin via the endothelin B receptor (EDNRB). The investigators used EDNRB KO cells to identify

specific phosphosites activated by endothelin treatment and build a signaling model using integration of prior knowledge kinase-substrate relationships and PPI with their time-resolved phosphoproteomics data. Using this approach endothelin activated kinases were predicted using PHOXTRACK and NetworKIN. Using the PHONEMeS approach a network was calculated resulting in five branching modules from the receptor (arrestin, Ca²⁺ depend kinases, PI3K-PDK1_AKT, PKC and PKA). The authors observe that even though the overlap of identified phosphosites between the two melanoma cells lines was not large, the structure and organization of the EDNRB signaling network was conserved. The authors validated the regulation of the identified major kinase nodes by WB and used inhibitors to assess the functional impact of EDNRB signaling in cell migration.

Overall the paper is well written and the data of high quality. The analysis strategy is valid and presents a novel insight in EDNRB signaling in melanoma cell lines. The reviewer recommends publication if one major concern can be addressed. The authors validated the predicted kinases using WB and propose that this and their inhibitor studies in cell migration assays validates the proposed network. However, it seems that these experiments do not address network topology. Inhibitor treatments with subsequent pSTY analysis could be used to validate predictions in the model. Will events downstream of inhibited kinases indeed be altered in pSTY in the way the network predicts? The reviewer recommends selection and inhibition of a few critical nodes with subsequent pSTY analysis and re-evaluation of the model.

Minor remarks:

- Why is the WB and pSTY data not shown for the A2058-EDNRB-KO cell line?
- Figure 2 B: Circles in Venn- Diagram do not reflect the actual size of the datasets e.g. proteins/phosphopeptides.

Reviewer #3:

Elucidating essential kinases of endothelin signalling by logic modelling of phosphoproteomics data by Alexander Schäfer et al. describes the phosphoproteomics analysis of endothelin (EDN) signalling, specifically focussing on endothelin B receptor activation in 2 melanoma cell lines. Using general phosphoproteomics they focussed on those phosphopeptide showing specific abundance changes strictly related to EDNRB activation. Then the authors used this dataset to predict the kinases involved in EDN signalling using PHOXTRACK (experimentally validated kinase-substrate relationships) and NetworKIN (predicted kinase-substrate relationships). Next, kinase and substrate data were combined to draw an EDNRB activated signalling network in the melanoma cell lines, followed by inhibitor studies and functional assays to validate the results.

Overall the manuscript is well written and adds relevant knowledge to the field of EDN signalling. There are some points that require addressing.

Major comment.

The authors perform several types of data analysis, ranging from PCA, kinase activity prediction, clustering, IPA, to a network modelling approach. It is however, unclear if and how all this data is integrated in the final modelling and whether any of the initial data analysis added useful information.

The initial data analysis seems less relevant, there is a section on PCA analysis, but the major discriminators are not revealed, nor what this actually means.

Two kinase prediction tools are used, however, there is little mention of the restrictions of such predictions, e.g. only well studied kinases, lot of redundancy, etc. Likely the PAK kinases are predicted from any dataset because of the broad range of biological processes these kinases are active in.

The exact process of the modelling approach is poorly described, except for a reference. Since a large part of the results of this study originate from the modelling it would be good to know what the input data exactly is and whether all the previous data analysis outputs are used.

Minor

Page 6. CREB phosphorylation seems more abundant at 10min in first plot.

Page 6. The description of the experimental setup is confusing. It seems like only cell line UACC257 is analysed in biological triplicate, the use of GIST is not immediately clear and Fig.1C doesn't clarify. This description can be improved.

Benefits of GIST are described, however, there is also the drawback of increased complexity, which limits number of identifications/quantifications. Indeed, the number of phosphopeptides in this study is relatively low, while enough studies use 'simply' label free approaches.

Page 14-15; the authors describe the overlap in the 2 cell lines being high in network topology even though the overlap in phosphopeptides is only 1/3. This is in line with the predicted active kinases. Again, this could be very well due to the fact that these pathways and kinases are well studied and thus can be predicted. It doesn't exclude differences between the cell lines in less well-known pathways.

Page 18. Discussion. The authors use of lot of text to highlight the quality of their data: "The breadth, specificity and quantitative accuracy of the obtained data", "comprehensive, robust phosphoproteomics datasets", "The high quality of the obtained time resolved global phosphorylation data". Although it's good to see the authors are pleased with their data, this section can be toned down a bit.

Along these lines, the statement on page 18 that: "The use of prior knowledge in the form of K-S relationships and PPIs, along with time resolved phosphoproteomic profiles, to build a dynamic, receptor activated signalling network is a new concept in signalling research." is very strong and according to this reviewer there are plenty of examples where, using different approaches, this is done (work of the Olsen group, publication on Photon, etc.)

Page 20, it's unclear what is meant with: "showing that kinase predictions are more consistent for a variety of perturbations than actual phosphorylation targets"

1st Revision - authors' response

17 May 2019

Reviewer #2:

The manuscript presented by Schäfer et al. is an interesting resource paper that provides data describing the phosphoproteomic landscape of melanoma cell lines after stimulation with endothelin via the endothelin B receptor (EDNRB). The investigators used EDNRB KO cells to identify specific phosphosites activated by endothelin treatment and build a signaling model using integration of prior knowledge kinase-substrate relationships and PPI with their time-resolved phosphoproteomics data. Using this approach endothelin activated kinases were predicted using PHOXTRACK and NetworKIN. Using the PHONEMeS approach a network was calculated resulting in five branching modules from the receptor (arrestin, Ca²⁺ depend kinases, PI3K-PDK1_AKT, PKC and PKA). The authors observe that even though the overlap of identified phosphosites between the two melanoma cells lines was not large, the structure and organization of the EDNRB signaling network was conserved. The authors validated the regulation of the identified major kinase nodes by WB and used inhibitors to assess the functional impact of EDNRB signaling in cell migration.

Overall the paper is well written and the data of high quality. The analysis strategy is valid and presents a novel insight in EDNRB signaling in melanoma cell lines.

We thank the reviewer for the positive comments

The reviewer recommends publication if one major concern can be addressed. The authors validated the predicted kinases using WB and propose that this and their inhibitor studies in cell migration assays validates the proposed network. However, it seems that these experiments do not address

network topology. Inhibitor treatments with subsequent pSTY analysis could be used to validate predictions in the model. Will events downstream of inhibited kinases indeed be altered in pSTY in the way the network predicts? The reviewer recommends selection and inhibition of a few critical nodes with subsequent pSTY analysis and re-evaluation of the model.

This is a very interesting suggestion to validate the model predictions, which we have followed and performed the required series of new experiments. The outcome is presented in Appendix Figure S2 and on page 16, 17 and 26. We selected MEK/ERK, PKC and p70S6K as central kinase nodes of the network for the inhibitor experiment. The impact of inhibition of these nodes on the effect of EDN stimulation at a single time point (20 min) was evaluated using quantitative phosphoproteomics. We found that the inhibitors blocked EDN induced phosphorylation of almost all direct and downstream kinase substrates predicted by our network model, confirming the structure of the model for all three kinase subnetworks tested.

Although this data is in accordance with our model, we also observed blocked phosphorylation of a few sites scattered across the network which were not directly linked to the target kinase in our model. Likewise, MEK/ERK and PKC inhibitors both inhibited p70S6K target site phosphorylation, indicating additional crosstalk between these modules which is currently not represented in our network model (and could not have been predicted since it was not part of the prior knowledge).

This is not surprising since the changes in phosphorylation following kinase inhibitors treatment may also result from (i) crossreactivity of the kinase inhibitor with unrelated kinases or (ii) indirect effects between inhibited kinase and substrate through intermediary kinases not included in the model or (iii) adaptive responses of inhibitor treatment (e.g. prevention of constitutive phosphorylations of unrelated kinases).

In addition, signalling pathways tend to be interlinked and inhibition of one major branch can affect the activity of other branches (e.g. RAF/MEK/ERK and PI3K/AKT; Mendoza, Trends Biochem Sci, 2011) even in the absence of ligand stimulation. Notwithstanding these nonlinear inhibitor responses, we believe that the results of the inhibitor experiments suggested by the reviewer added important new data for validating the predicted network structure in the three tested kinase subnetworks and thus strengthen our EDN signalling network.

Minor remarks:

- Why is the WB and pSTY data not shown for the A2058-EDNRB-KO cell line?

These experiments were not performed on the A2058 EDNRB-KO cell line. We decided to focus mainly on the more melanocytic UACC257 cell line and used data from A2058 cells as confirmatory evidence. The A2058-EDNRB-KO experiments could be removed from the manuscript entirely but we found it helpful to show that EDN also signals through a single receptor in a second melanoma cell line.

- Figure 2 B: Circles in Venn- Diagram do not reflect the actual size of the datasets e.g. proteins/phosphopeptides.

Figure 2B has been changed accordingly and the circles now reflect the dataset sizes.

Reviewer #3:

Elucidating essential kinases of endothelin signalling by logic modelling of phosphoproteomics data by Alexander Schäfer et al. describes the phosphoproteomics analysis of endothelin (EDN) signalling, specifically focussing on endothelin B receptor activation in 2 melanoma cell lines. Using general phosphoproteomics they focussed on those phosphopeptide showing specific abundance changes strictly related to EDNRB activation. Then the authors used this dataset to predict the kinases involved in EDN signalling using PHOXTRACK (experimentally validated kinase-substrate relationships) and NetworKIN (predicted kinase-substrate relationships). Next, kinase and substrate data were combined to draw an EDNRB activated signalling network in the melanoma cell lines, followed by inhibitor studies and functional assays to validate the results.

Overall the manuscript is well written and adds relevant knowledge to the field of EDN signalling. There are some points that require addressing.

We thank the reviewer for his positive assessment of our study

Major comment.

The authors perform several types of data analysis, ranging from PCA, kinase activity prediction, clustering, IPA, to a network modelling approach. It is however, unclear if and how all this data is integrated in the final modelling and whether any of the initial data analysis added useful information.

All analyses were run in parallel and none of the mentioned analyses were integrated in the network model. This is now explicitly stated in the modelling section (page 13, paragraph 2). All mentioned analyses did, however, contribute useful information to flesh out the analysis of our phosphoproteomic dataset. Logical modelling generated the most impressive results, but it was limited by the availability of prior knowledge. PCA, IPA and kinase activation prediction contribute biological insights which could not be integrated in the logical models.

The initial data analysis seems less relevant, there is a section on PCA analysis, but the major discriminators are not revealed, nor what this actually means.

The PCA analysis showed that biological replicates cluster together, demonstrating the quality of the data. We also observed separation of the samples based on the main biological variables (expression and activation of EDNRB), showing that these variables translate to phosphorylation pattern differences. The conclusions are now stated at the end of the PCA section (page 8, paragraph 1).

Two kinase prediction tools are used, however, there is little mention of the restrictions of such predictions, e.g. only well studied kinases, lot of redundancy, etc. Likely the PAK kinases are predicted from any dataset because of the broad range of biological processes these kinases are active in.

We completely agree that the K-S relationship prior knowledge has significant shortcomings in terms of redundancy and their focus on well-studied signalling pathways. Both kinase prediction methods use statistical procedures, (e.g. Kolmogorov–Smirnov enrichment statistics for PHOXTRACK) to guard against the influence of substrate frequency on their predictions.

Kinase activation prediction was an important complementary analysis to the network model because it contains checks against substrate frequency in the prior knowledge, which the modelling approach did not have. It was therefore encouraging that both network modelling and kinase activation prediction resulted in very similar sets of EDN target kinases. PAK activation by EDNRB in particular was validated by WB in Figure 6.

The exact process of the modelling approach is poorly described, except for a reference. Since a large part of the results of this study originate from the modelling it would be good to know what the input data exactly is and whether all the previous data analysis outputs are used.

This point is well taken and we have expanded the Methods section with a detailed explanation of the modelling procedure and input data (page 31, 32). Computer code in R for the modelling procedure and input data has been submitted to Github (https://github.com/saezlab/EDN_phospho). Input data consisted of the quantitative phosphosite tables (Table EV1b and EV2b, part of the manuscript) and prior knowledge as a list of binary kinase substrate relationships and protein-proteins interactions, derived from Omnipath.

Minor

Page 6. CREB phosphorylation seems more abundant at 10min in first plot.

This mistake has been corrected (page 5, paragraph 3). “On the phosphorylation level, EDN caused a transient induction of CREB S133 phosphorylation (Fig. 1B) with a maximum at 2 min in both cell lines”

Has been changed to

“On the phosphorylation level, EDN caused a transient induction of CREB S133 phosphorylation (Fig. 1B) with a maximum at 10 min in UACC257 and 2 min in A2058”

Page 6. The description of the experimental setup is confusing. It seems like only cell line UACC257 is analysed in biological triplicate, the use of GIST is not immediately clear and Fig.1C doesn't clarify. This description can be improved.

The legend for Figure 1C has been expanded to clarify the questions raised by the reviewer The A2058 dataset also contains biological triplicates. The following has been added to the legend of Figure 1C (page 36, paragraph 3). “The GIST was generated by pooling all heavy SILAC plates (blue) and spiked into all UACC257 (red) and UACC257 EDNRB-KO (green) samples. The A2058 cell line was also analysed in biological triplicates and a GIST but without EDNRB-KO cells.”

Benefits of GIST are described, however, there is also the drawback of increased complexity, which limits number of identifications/quantifications. Indeed, the number of phosphopeptides in this study is relatively low, while enough studies use 'simply' label free approaches.

We agree with this comment and have added the caveat to the GIST description. The following sentence has been added on page 7, paragraph 2. “However, the GIST approach shares a major drawback of SILAC - a reduction of identification rates due to increased sample complexity.”

Page 14-15; the authors describe the overlap in the 2 cell lines being high in network topology even though the overlap in phosphopeptides is only 1/3. This is in line with the predicted active kinases. Again, this could be very well due to the fact that these pathways and kinases are well studied and thus can be predicted. It doesn't exclude differences between the cell lines in less well-known pathways.

This point is well taken. Kinase activation prediction and network modelling rely on publicly available prior knowledge, which is biased in favour of well-studied pathways. This is not due to a lack of specificity of the predictions, as the Western blot validations and networks generated with scrambled input data shows. But there may well be differences in pathways that are not represented in the prior knowledge and are thus not present in our analysis. This is now mentioned in the respective section (page 15, paragraph 1):

“However, it cannot be excluded that less well-studied pathways are different between the two cell lines, since kinase activation prediction and network modelling rely on publicly available prior knowledge which is biased in favour of well-studied signalling pathways.”

Page 18. Discussion. The authors use of lot of text to highlight the quality of their data: "The breadth, specificity and quantitative accuracy of the obtained data", "comprehensive, robust phosphoproteomics datasets", "The high quality of the obtained time resolved global phosphorylation data". Although it's good to see the authors are pleased with their data, this section can be toned down a bit.

This section has been toned down accordingly.

Along these lines, the statement on page 18 that: "The use of prior knowledge in the form of K-S relationships and PPIs, along with time resolved phosphoproteomic profiles, to build a dynamic, receptor activated signalling network is a new concept in signalling research." is very strong and according to this reviewer there are plenty of examples where, using different approaches, this is done (work of the Olsen group, publication on Photon, etc.)

We have removed the cited paragraph from the discussion.

Page 20, it's unclear what is meant with: "showing that kinase predictions are more consistent for a variety of perturbations than actual phosphorylation targets"

This statement on page 20 has been clarified. It now reads:

“showing that the same cell perturbations in different studies led to the same kinase activation predictions, although the actual phosphopeptides underlying these predictions were different”

2nd Editorial Decision

4 July 2019

Thank you for sending us your revised manuscript. We have now heard back from the two reviewers who were asked to evaluate your study. As you will see the reviewers are satisfied with the modifications made and think that the study is now suitable for publication after completing minor editorial amendments.

REFEREE REPORTS

Reviewer #2:

The authors have addressed all of my concerns and present an important phosphoproteomic resource of EDN signaling.

Reviewer #3:

All comments have been addressed and the manuscript can be published

2nd Revision - authors' response

9 July 2019

The authors made the requested editorial changes.

Corresponding Author Name: Alexander Schäfer

Manuscript Number: MSB-19-8828